# The impact of chlorine chemistry combined with heterogeneous $N_2O_5$ reactions on air quality in China

Xiajie Yang[1,2], Qiaoqiao Wang[1,2], Nan Ma[1,2], Weiwei Hu[3], Yang Gao[4], Zhijiong Huang[1,2], Junyu Zheng[1,2], Bin Yuan[1,2], Ning Yang[1,2], Jiangchuan Tao[1,2], Juan Hong[1,2], Yafang Cheng[5], Hang Su[5]

[1] Institute for Environmental and Climate Research, Jinan University, Guangzhou 511443, China.

[2] Guangdong-Hongkong-Macau Joint Laboratory of Collaborative Innovation for Environmental Quality, Guangzhou 511443, China.

[3] Guangzhou Institute of Geochemistry, Chinese Academy of Sciences, Guangzhou 510640, China.

[4] Key Laboratory of Marine Environment and Ecology, Ministry of Education, Ocean University of China, Qingdao 266100, China

[5] Max Planck Institute for Chemistry, Mainz 55128, Germany

*Correspondence to*: Qiaoqiao Wang (qwang@jnu.edu.cn)

**Abstract:** The heterogeneous reaction of $N_2O_5$ on Cl-containing aerosols (heterogeneous $N_2O_5$ + Cl chemistry) plays a key role in chlorine activation, $NO_x$ recycling and consequently $O_3$ and $PM_{2.5}$ formation. In this study, we use the GEOS-Chem model with additional anthropogenic and biomass burning chlorine emissions combined with updated parameterizations for the heterogeneous $N_2O_5$ + Cl chemistry (i.e. the uptake coefficient of $N_2O_5$ ($\gamma_{N2O5}$) and the $ClNO_2$ yield ($\varphi_{ClNO2}$)) to investigate the impacts of chlorine chemistry on air quality in China, the role of the heterogeneous $N_2O_5$ + Cl chemistry, as well as the sensitivity of air pollution formation to chlorine emissions and parameterizations for $\gamma_{N2O5}$ and $\varphi_{ClNO2}$. The model simulations are evaluated against multiple observational datasets across China and show significant improvement in reproducing observations of particulate chloride, $N_2O_5$, and $ClNO_2$ when including anthropogenic chlorine emissions and updates to the parameterization of the heterogeneous $N_2O_5$ + Cl chemistry relative to the default model. The simulations show that total tropospheric chlorine chemistry could increase annual mean maximum daily 8-hour average (MDA8) $O_3$ by up to 4.5 ppbv but decrease $PM_{2.5}$ by up to 7.9 μg m$^{-3}$ in China, 83% and 90% of which could be attributed to the effect of the heterogeneous $N_2O_5$ + Cl chemistry. The heterogeneous uptake of $N_2O_5$ on chloride-containing aerosol surfaces is an important loss pathway of $N_2O_5$ as well as an important source of $O_3$, and hence is particularly useful in elucidating the commonly seen ozone underestimations relative to observations. The importance of chlorine chemistry largely depends on both chlorine emissions and the parameterizations for the heterogeneous $N_2O_5$ + Cl chemistry. With the additional chlorine emissions, the simulations show that annual MDA8 $O_3$ in China could be increased by up to 3.5 ppbv. The corresponding effect on $PM_{2.5}$ concentrations varies largely with regions, with an increase of up to 4.5 μg m$^{-3}$ in the North China Plain but a decrease of up to 3.7 μg m$^{-3}$ in the Sichuan Basin. On the other hand, even with the same chlorine emissions, the effects on MDA8 $O_3$ and $PM_{2.5}$ in China could differ by 48% and 27%, respectively between different parameterizations.

 **1 Introduction**

Chlorine (Cl) plays an important role in atmospheric chemistry in both the stratosphere and the
troposphere, primarily via the reactions of Cl atom with various atmospheric trace gases including
dimethyl sulfide, methane, and other volatile organic compounds (VOCs). The chemistry of Cl is quite
similar with that of hydroxyl radicals (OH) while Cl atom reacts up to 2 orders of magnitude faster with
some VOCs than OH (Atkinson et al., 2006). Studies have shown that Cl accounts for around 2.5% –
2.7% of the global $CH_4$ oxidation in the troposphere, and the contribution varies across regions, reaching
up to 10% – 15% in Cape Verde and ~20% in east China (Lawler et al., 2011; Hossaini et al., 2016). Cl
atom, therefore, is regarded as a potentially important tropospheric oxidant.
In general, Cl atom can be produced from the photo-dissociation and the oxidation of chlorinated
organic species (e.g. $CH_3Cl$, $CH_2Cl_2$ and $CHCl_3$) and inorganic chlorine species (i.e. HCl and $Cl_2$)
(Saiz-Lopez and Von Glasow, 2012; Simpson et al., 2015). Nitryl chloride ($ClNO_2$), formed through the
heterogeneous reaction between dinitrogen pentoxide ($N_2O_5$) and chloride-containing aerosols
(hereinafter referred to as the heterogeneous $N_2O_5$ + Cl chemistry), is found to be another important
source of tropospheric Cl atoms in polluted regions (Liu et al., 2018; Haskins et al., 2019; Choi et al.,
2020; Simpson et al., 2015). The heterogeneous formation of $ClNO_2$ and the subsequent photolysis can
be described by reactions R1 – R4 shown below (Finlayson-Pitts et al., 1989; Osthoff et al., 2008). The
net reactions of R1 ($N_2O_5$ hydrolysis on none-chloride-containing aerosols) and R2 (uptake of $N_2O_5$ on
chloride-containing aerosols) could be expressed as R3, in which the $ClNO_2$ yield (i.e. $\varphi_{ClNO2}$, defined
as the molar ratio of produced $ClNO_2$ to total reacted $N_2O_5$) represents the fraction of $N_2O_5$ reacting via
R2.
$N_2O_5\,(g) + H_2O\,(aq) \rightarrow 2\,HNO_3\,(aq)$ (R1)
$N_2O_5\,(g) + HCl\,(aq) \rightarrow HNO_3\,(aq) + ClNO_2\,(g)$ (R2)
$N_2O_5\,(g) + (1-\varphi)\,H_2O\,(aq) + \varphi\,HCl\,(aq) \rightarrow (2-\varphi)\,HNO_3\,(aq) + \varphi\,ClNO_2\,(g)$ (R3)
$ClNO_2\,(g) + hv \rightarrow Cl\,(g) + NO_2\,(g)$ (R4)
Estimates based on model simulations have suggested that $ClNO_2$ provides a source of Cl atoms totaling
0.66 Tg Cl a$^{-1}$, with the vast majority (95%) being in the Northern Hemisphere (Sherwen et al., 2016).
The relative contribution of $ClNO_2$ to global tropospheric Cl atoms is 14% on average and exhibits clear
regional variations (Sherwen et al., 2016). For example, the study by Riedel et al. (2012) reported that
the relative contribution is approximately 45% in Los Angeles based on a simple box model combined
with local observations.
The heterogeneous formation of $ClNO_2$ also serves as a reservoir for reactive nitrogen at night. The rapid
photolysis of $ClNO_2$ at daytime (R4) not only releases highly reactive Cl atom but also recycles $NO_2$ back
to the atmosphere, which as well significantly affect the daytime photochemistry (Riedel et al., 2014).
Previous global and hemispheric models found that the heterogeneous $N_2O_5 + Cl$ chemistry could
increase monthly mean values of the maximum daily 8h average (MDA8) $O_3$ concentrations by $1.0 - 8.0$
ppbv in most Northern Hemisphere regions (Sarwar et al., 2014; Wang et al., 2019). The reaction also
impacts secondary aerosol formation, mainly through the recycling of $NO_x$ (Staudt et al., 2019; Mitroo
et al., 2019). For example, Sarwar et al. (2014) estimated that $ClNO_2$ production decreases nitrate by 3.3%
in winter and 0.3% in summer averaged over the entire Northern Hemisphere. The influence of the
heterogeneous formation of $ClNO_2$ in China is even larger due to the polluted environment, leading to an
increase in ozone concentrations by up to 7 ppbv, and a decrease in total nitrate by up to 2.35 µg m$^{-3}$ on
monthly mean basis (Li et al., 2016; Sarwar et al., 2014)
There are two key parameters that determine the uptake efficiency of $N_2O_5$ and production of $ClNO_2$, the
aerosol uptake coefficient of $N_2O_5$ ($\gamma_{N2O5}$) and the ClNO2 yield ($\varphi_{ClNO2}$). The most widely used
parameterization for $\gamma_{N2O5}$ and $\varphi_{ClNO2}$ was proposed by Bertram and Thornton (2009) (hereinafter referred
to as BT09), which is based on the laboratory studies with considerations of aerosol water content,
concentrations of nitrate and chloride, and specific surface area (i.e. the ratio of surface area
concentrations to particle volume concentrations). However, recent field and model studies have shown
that this parameterization would overestimate both $\gamma_{N2O5}$ and $\varphi_{ClNO2,}$ especially in regions with high Cl
levels (Mcduffie et al., 2018b; Mcduffie et al., 2018a; Xia et al., 2019; Chang et al., 2016; Hong et al.,
2020; Yu et al., 2020). The discrepancies could be partly attributed to the complexity of atmospheric
aerosols (e.g. mixing state and complex coating materials) in contrast to the simple proxies used in
laboratory studies (Yu et al., 2020). Specifically, the suppressive effect of organic coatings is not
considered in BT09. Several parameterizations updated from BT09 have been proposed by more recent
studies based on field measurements and box model studies (Yu et al., 2020; Mcduffie et al., 2018a;
Mcduffie et al., 2018b; Xia et al., 2019). However, some of these previous field-based parameterizations
were derived from observations under different ambient conditions which may not be applicable to the
highly polluted regions in China. A full evaluation of the representativeness of different
parameterizations for the heterogeneous $N_2O_5 + Cl$ chemistry and the associated impacts on ambient air
quality in China is not available yet.
In addition to the parameterization, the influence of the heterogeneous $N_2O_5 + Cl$ chemistry is also
sensitive to chlorine emissions. In early modelling studies, global tropospheric chlorine is mainly from
sea salt aerosols (SSA), and most of the chlorine over continental regions in North America and Europe
is dominated by the long-range transport of SSA (Wang et al., 2019; Sherwen et al., 2017). The study by
Wang et al. (2019) found an addition of anthropogenic chlorine emissions in the model would result in
overestimates of HCl observations in the U.S and suggested insignificant influence of anthropogenic Cl
in the U.S. However, there are also studies pointing out the importance of anthropogenic chlorine
emissions in China (Le Breton et al., 2018; Yang et al., 2018; Hong et al., 2020). The study by Wang et
al. (2020b) suggested that anthropogenic chlorine emissions in China are more than 8 times higher than
those in the U.S., and could dominate reactive chlorine in China, resulting in an increase in $PM_{2.5}$ and
Ozone by up to 3.2 μg m$^{-3}$ and 1.9 ppbv on annual mean basis, respectively. The comprehensive effects
of anthropogenic chlorine on air quality as well as their sensitivities to different parameterizations for
the heterogeneous $N_2O_5 + Cl$ chemistry, however, has not been investigated in previous studies.
In this work, we use the GEOS-Chem model to investigate the impacts of chlorine chemistry including
the heterogeneous $N_2O_5 + Cl$ chemistry on air quality in China. Multiple observational data sets,
including $N_2O_5$, $ClNO_2$, $O_3$, $PM_{2.5}$ and its chemical compositions from different representative sites
across China, are used to assess the model performance. With comprehensive chlorine emissions as well
as appropriate parameterizations for the heterogeneous $N_2O_5 + Cl$ chemistry, our objectives are: 1) to
improve the model's performance regarding the simulation of particulate chloride, $ClNO_2$, $N_2O_5$, $PM_{2.5}$
and $O_3$ concentrations; and 2) to extend the investigation on the effects of chlorine chemistry on both
$PM_{2.5}$ and ozone pollution in China as well as their sensitivities to anthropogenic chlorine emissions and
the parameterizations for the heterogeneous $N_2O_5$ + Cl chemistry.

## 2 Methodology

### 2.1 GEOS-Chem Model

The GEOS-Chem model (version 12.9.3, http://www.geos-chem.org, DOI: 10.5281/zenodo.3974569) is
driven by assimilated meteorological fields GEOS-FP from the NASA Global Modeling and Assimilation
Office (GMAO) at NASA Goddard Space Flight Center. The simulation in this study was conducted in
a nested-grid model with a native horizontal resolution of $0.25° \times 0.3125°$ (latitude $\times$ longitude) and 47
vertical levels over East Asia (70° − 140° E, 15° S − 55° N). The dynamical boundary conditions were
from a global simulation with a horizontal resolution of $2° \times 2.5°$. We initialized the model with a 1-month
spin up followed by a 1-year simulation for 2018. The simulation included a detailed representation of
coupled $NO_x$ – ozone – VOCs – aerosol − halogen chemistry (Sherwen et al., 2016; Wang et al., 2019).
Previous studies have demonstrated the ability of GEOS-Chem to reasonably reproduce the magnitude
and seasonal variation of surface ozone and particulate matter over East Asia and China (Wang et al.,
2013; Geng et al., 2015; Li et al., 2019).

### 2.1.1 Chlorine Chemistry

The GEOS-Chem model includes a comprehensive chlorine chemistry mechanism coupled with bromine
and iodine chemistry. Full details could be found in the study of Wang et al. (2019). Briefly, the model
includes 12 gas-phase inorganic chlorine species (Cl, $Cl_2$, $Cl_2O_2$, $ClNO_2$, $ClNO_3$, ClO, ClOO, OClO,
BrCl, ICl, HOCl, HCl), 3 gas-phase organic chlorine ($CH_3Cl$, $CH_2Cl_2$, $CHCl_3$), and aerosol $Cl^-$ in two
size bins (fine mode with radius $\leqslant$ 0.5 µm and coarse mode with radius > 0.5 µm). The gas-aerosol
equilibrium between HCl and $Cl^-$ is calculated with ISORROPIA II (Fountoukis and Nenes, 2007) as
part of the $H_2SO_4$ – HCl − $HNO_3$ − $NH_3$ – non-volatile cations (NVCs) thermodynamic system, where
$Na^+$ is used as a proxy for NVCs.
The heterogeneous uptake of $N_2O_5$ on aerosol surfaces leading to the production of $ClNO_2$ and $HNO_3$
has also been included in GEOS-Chem with the parameterizations for $\gamma_{N2O5}$ and $\varphi_{ClNO2}$ proposed by
McDuffie et al. (2018b; 2018a) by default (hereinafter referred to as McDuffie parameterization).
McDuffie parameterization is the first field-based empirical parameterization derived from the
framework proposed in multiple laboratory studies including BT09 (Anttila et al., 2006; Bertram and
Thornton, 2009; Riemer et al., 2009) to account for the uptake dependence on aerosol water and nitrate
concentrations as well as the resistance from an organic coating. The coefficients for McDuffie
parameterization were derived from applying a box model to observations of $N_2O_5$, $ClNO_2$, $O_3$, and $NO_x$
mixing ratios during the winter in the eastern U.S. The parameterization for $\gamma_{N2O5}$ accounts for both the
inorganic and organic aerosol components and can be described by Eq. $1-3$:
$$\frac{1}{\gamma_{N_2O_5}} = \frac{1}{\gamma_{core}} + \frac{1}{\gamma_{coat}} \qquad \text{Eq. 1}$$

$$\gamma_{core} = \frac{4V}{c \cdot S_a} K_H \times 2.14 \times 10^5 \times [H_2O]\left(1 - \frac{1}{k_a \frac{[H_2O]}{[NO_3^-]} + 1}\right) \qquad \text{Eq. 2}$$

$$\gamma_{coat} = \frac{4RT\varepsilon H_{aq}D_{aq}R_c}{clR_p} \qquad \text{Eq. 3}$$

Where $\gamma_{core}$ represents the reactive uptake of inorganic aerosol core and $\gamma_{coat}$ represents the retardation of
the organic coating; $c$ is the average gas-phase thermal velocity of $N_2O_5$ (m s$^{-1}$), $V$ is the total particle
volume concentration (m$^3$ m$^{-3}$), $S_a$ is the total particle surface area concentration (m$^2$ m$^{-3}$), $K_H$ is the
unitless Henry's law coefficient for $N_2O_5$ with a constant value of 51, $[H_2O]$ and $[NO_3^-]$ are the
concentrations of aerosol liquid water content and aerosol nitrate (mol L$^{-1}$), respectively, and $k_a$ is the
rate constant ratio representing the competition between aerosol-phase $H_2O$ and $NO_3^-$ for the
$H_2ONO_2^+$(aq) intermediate and is fixed at 0.04 in Eq.2; $R$ is the ideal gas constant, $T$ is the temperature
(K), $H_{aq}$ and $D_{aq}$ are the aqueous Henry's law constant and aqueous-phase diffusion coefficient of $N_2O_5$,
respectively, $\varepsilon$ is a linear combination of relative humidity (RH) and O:C ratio (= 0.15 × O:C + 0.0016 ×
RH), and $R_p$, $R_c$, and $l$ are the total particle radius, inorganic core radius and organic coating thickness,
respectively (m).
$\varphi_{ClNO2}$ is calculated following BT09, but is reduced by 75% based on the observations conducted in
eastern U.S. and offshore in spring 2015 (i.e. the WINTER aircraft campaign) (McDuffie et al., 2018b).
It could be described by Eq. 4:
$$\varphi_{ClNO_2} = 0.25 \times \left(k_c \frac{[H_2O]}{[Cl^-]} + 1\right)^{-1} \qquad \text{Eq. 4}$$

Where $k_c$ is the rate constant ratio representing the competition between aerosol-phase $H_2O$ and $Cl^-$ for
the $H_2ONO_2^+$(aq) intermediate and is fixed at 1/450 here, and [$Cl^-$] is the concentration of aerosol chloride
(mol $L^{-1}$). For more detailed description of McDuffie parameterization, readers are referred to McDuffie
et al. (2018b; 2018a). Keep it in mind that the coefficients for the parameterizations in Eq. 1 – 4 were
derived to better reproduce wintertime observations in the eastern U.S. However, there are large
uncertainties in both the values of the coefficients and functional form of the parameterizations,
specifically related to their applicability to other regions.
Recently, Yu et al. (2020) proposed new parameterizations of $\gamma_{N2O5}$ and $\varphi_{ClNO2}$ based on BT09 to account
for the dependence on aerosol water, nitrate, and chloride concentrations but with coefficients derived
from uptake coefficients directly measured on ambient aerosol in two rural sites in China. The
parameterizations of $\gamma_{N2O5}$ and $\varphi_{ClNO2}$ (hereinafter referred to as Yu parameterization) are described by
Eq. 5 – 6, respectively:
$$\gamma_{N_2O_5} = \frac{4V}{c \cdot S_a} K_H \times 3.0 \times 10^4 \times [H_2O] \left( 1 - \frac{1}{k_a \frac{[H_2O]}{[NO_3^-]} + k_b \frac{[Cl^-]}{[NO_3^-]} + 1} \right) \quad \text{Eq. 5}$$

$$\varphi_{ClNO_2} = \left( 1 + k_c \frac{[H_2O]}{[Cl^-]} \right)^{-1} \quad\quad\quad \text{Eq. 6}$$

Where $k_b$ is the rate constant ratio representing the competition between aerosol-phase $Cl^-$ and $NO_3^-$ for
the $H_2ONO_2^+$(aq) intermediate and is fixed at 3.4. In contrast to McDuffie parameterization, $k_a$ and $k_c$ in
Yu parameterization are fixed at 0.033 and 1/150, respectively.
Although both the two parameterizations are developed based on BT09, there exit significant differences
of $\gamma_{N2O5}$ and $\varphi_{ClNO2}$ between McDuffie and Yu parameterizations. For $\gamma_{N2O5}$, McDuffie parameterization
generally follows BT09 for the calculation of the uptake on inorganic aerosols (i.e. $\gamma_{core}$), but excludes
the dependence on aerosol chloride so as to better reproduce observed wintertime reactive nitrogen in
eastern U.S. Moreover, the parameterization accounts for the suppressive effects of the organics (i.e. $\gamma_{coat}$),
which is not directly included in BT09 (Anttila et al., 2006; Riemer et al., 2009; Morgan et al., 2015). In
contrast to McDuffie parameterization, Yu parameterization excludes the organic suppression but
includes the chloride enhancement so as to better reproduce $\gamma_{N2O5}$ observed in China (Yu et al., 2020). It
is worth mentioning that the coefficients applied in the parameterization of $\gamma_{N2O5}$ also differ between
McDuffie and Yu parameterizations as both are fixed to reproduce the ambient observation representing
different pollution conditions. For example, $k_a$ is equal to 0.04 in Eq. 2 but 0.033 in Eq. 5. The $\gamma_{N2O5}$ in
McDuffie parameterization is thus expected to be lower compared with the Yu parameterization due to
the resistance from organic coating and the lack of the chloride enhancement. For $\varphi_{ClNO2}$, both the
McDuffie and Yu parameterizations are based on BT09, but with different coefficients (i.e. $k_c = 1/450$ in
Eq. 4 and 1/150 in Eq. 6). Although $k_c$ in Eq. 4 is relatively smaller, the scaling factor of 0.25 applied in
Eq. 4 ultimately results in a much smaller $\varphi_{ClNO2}$ in McDuffie parameterization compared with Yu
parameterization under the same condition. Again, keep it in mind that McDuffie parameterization is
derived from fits to observations over the eastern U.S. (McDuffie et al., 2018a) while Yu parameterization
is fitted to observations at rural locations in China (Yu et al., 2020).
In this study, we updated the parameterizations for $\gamma_{N2O5}$ and $\varphi_{ClNO2}$ in the heterogeneous $N_2O_5$ + Cl
chemistry (hereinafter referred to as parameterizations for heterogeneous $N_2O_5$ + Cl chemistry) in the
GEOS-Chem with Yu parameterization. Additional simulation cases were also performed to evaluate the
representativeness of both Yu and McDuffie parameterizations regarding the simulation of $N_2O_5$, $ClNO_2$,
$O_3$, $PM_{2.5}$ and its chemical compositions in China. Detailed description of the model setup for related
cases is provided below in Section 2.1.3.

**2.1.2 Emissions**

The study uses the Hemispheric Transport of Air Pollution (HTAPv2, http://www.htap.org/) based on the
emission of 2010 as a global anthropogenic inventory. This inventory is overwritten by a regional
emission inventory MIX (with a horizontal resolution of $0.25° \times 0.25°$) over East Asia based on the
emission in 2017, which is developed for the Model Inter-Comparison Study for Asia (MICS-Asia) and
covers all major anthropogenic sources in 30 Asian countries and regions (Li et al., 2017). In addition,
anthropogenic emissions of black carbon (BC) and organic carbon (OC) in Guangdong province, China
$(109° - 117°$ E, $20° - 26°$ N) are overwritten by a more recent high-resolution inventory (9 km $\times$ 9 km)
described by Huang et al. (2021). Biomass burning emissions are from the Global Fire Emissions
Database (GFED4) (Van et al., 2010) with a 3-hour time resolution. And the biogenic emissions of VOCs
are calculated based on the Model of Emissions of Gases and Aerosols from Nature (MEGAN2.1)
(Guenther et al., 2006).
Table 1 lists Cl emissions from all sources in the model. The global tropospheric chlorine by default in
the model is mainly from the mobilization of Cl$^-$ from SSA distributed over two size bins (fine and coarse
modes) (Wang et al., 2019), which is computed online as the integrals of the size-dependent source
function depending on wind speeds and sea surface temperatures (Jaeglé et al., 2011). During the
simulation year of 2018, SSA contributes $6.5 \times 10^4$ Gg Cl$^-$, most of which however are distributed over
the ocean due to its relatively short lifetime (~1.5 days) (Choi et al., 2020). The release of atomic Cl
from organic chlorine ($CH_3Cl$, $CH_2Cl_2$ and $CHCl_3$) via the oxidation by OH and Cl is also included in
the model by default. These organic chlorine gases are mainly of biogenic marine origin (Simmonds et
al., 2006), with a mean tropospheric lifetime longer than 250 days (Wang et al., 2020b), and are simulated
in the model by imposing fixed surface concentrations as described by Schmidt et al. (2016). Total
emissions of Cl atom from $CH_3Cl$, $CH_2Cl_2$, and $CHCl_3$ are calculated to be 3.8, 2.4, and 0.70 Gg Cl a$^{-1}$,
respectively.
Considering the importance of anthropogenic chlorine in China, we have further updated chlorine
inventories in the model to account for anthropogenic HCl, $Cl_2$ and fine particulate Cl$^-$, as well as biomass
burning HCl and Cl$^-$ emissions (also shown in Table 1 and S1). For fine particulate Cl$^-$ from both
anthropogenic and biomass burning, the emissions are estimated based on $PM_{2.5}$ emissions from MIX
and GFED4 inventories combined with the emission ratios of fine particulate Cl$^-$ versus $PM_{2.5}$ for
different emission sectors adopted from the study of Fu et al. (2018). Estimated Cl$^-$ emissions from
anthropogenic and biomass burning are 379 and 120 Gg, respectively, comparable to the results of 486
Gg in total for the year of 2014 by Fu et al. (2018). The anthropogenic emissions of HCl and $Cl_2$ are from
ACEIC (Anthropogenic Chlorine Emissions Inventory for China) (Liu et al., 2018) and are estimated to
be 218 and 8.9 Gg Cl in China, respectively. For HCl from biomass burning, the emission factors from
Lobert et al. (1999) are used for different types of biomass provided in GFED4, and a total emission of
30 Gg Cl is obtained in China in 2018. Total implemented chlorine emission for the simulation year of
2018 is 756 Gg Cl.
Figure 1 shows the distribution of Cl emissions from the sources mentioned above. Anthropogenic and
biomass burning emissions of HCl are concentrated in the Northeast Plain, North China Plain, Yangtze
River Delta, and Sichuan Basin, and are up to 320 kg Cl km$^{-2}$ a$^{-1}$ in the Sichuan Basin. Emissions of HCl
are low in South China, mainly due to the low chlorine content of coal in these regions (Hong et al.,
2020). The relative contribution of biomass burning to total HCl emissions in China is 14% on average
but could become dominant in the Northeast Plain due to the discrepancies in the spatial distributions of
anthropogenic and biomass burning emissions. The anthropogenic $Cl_2$ emissions have a similar spatial
distribution to that of HCl, but are one order of magnitude lower than HCl emissions. The distribution of
non-sea salt $Cl^-$ emissions is also similar to that of HCl and $Cl_2$, except that non-sea salt $Cl^-$ emissions
are also high in Central China. In contrast, emissions of sea salt $Cl^-$ (Fig. S1) are mainly distributed over
the ocean, implying limited influences over inland due to rapid deposition during transport. The spatial
distributions of different organic chlorine sources are similar, with maximums ($\sim 0.5$ kg Cl km$^{-2}$ a$^{-1}$) in
coastal regions (Fig. S1).

### 2.1.3 Model setup for different simulation cases


In this study, we conducted a series of simulation cases to investigate the effects of chlorine chemistry
on air quality in China, the role of $N_2O_5 - ClNO_2$ chemistry, and the associated sensitivities to chlorine
emissions as well as the parameterizations for $N_2O_5 - ClNO_2$ chemistry. Detailed model setup for those
cases is listed in Table 2. The Base case is the one with all updates in this study, including additional
chlorine sources from anthropogenic and biomass burning emissions as well as $N_2O_5$ uptake and $ClNO_2$
production represented by Yu parameterization. The NoEm case is conducted with a similar setup as the
Base case but only includes chlorine emissions from SSA and organic chlorine sources so as to evaluate
the model improvement originated from the updated chlorine emissions through the comparison with the
Base case. The McDuffie case is also performed using the McDuffie instead of Yu parameterization for
$\gamma_{N2O5}$ and $\varphi_{ClNO2}$ while keeping others the same as the Base case so as to evaluate the discrepancies
originated from different parameterizations for the heterogeneous $N_2O_5 + Cl$ chemistry.
In addition, while keeping others the same as the Base case, the NoHet case sets $\varphi_{ClNO2}$ to zero (Eq.6)
and removes the enhancement of $N_2O_5$ uptake from aerosol chloride (i.e. $[Cl^-] = 0$ in Eq. 5). The
comparison between the Base and NoHet cases could thus evaluate the importance of the heterogeneous
$N_2O_5 + Cl$ chemistry (i.e., the model sensitivities to a smaller gamma $N_2O_5$ and zero $ClNO_2$ production).
Similarly, combined with three more sensitivity cases (NoChem, NoEmHet and NoAll, see details in
Table 2), the study provides an overall evaluation of the importance of tropospheric chlorine chemistry
as well as its sensitivities to chlorine emissions and the parameterizations for the heterogeneous $N_2O_5 +$
Cl chemistry in the model.
**2.2 Observations**
Multiple observed data sets were applied in this study to evaluate the performance of GEOS-Chem
simulation, including the concentrations of chemical compositions of $PM_{2.5}$ from three representative
sites, located in south (Guangzhou, 23.14° N, 113.36° E), east (Dongying, 37.82° N, 119.05° E) and north
(Gucheng, 37.36° N, 115.96° E) China, respectively (Fig. S2). Concentrations of $SO_4^{2-}$, $NO_3^-$, $NH_4^+$, $Cl^-$
and organic matter (OM) in $PM_{2.5}$ were measured by High Resolution Time-of-Flight Aerosol Mass
Spectrometer (HR-ToF-AMS; Aerodyne Research Inc., USA, Decarlo et al. (2006)) from October 2 to
November 18, 2018 (with a time resolution of 1 minute) at Guangzhou site (Chen et al., 2021b), and
from March 18 to April 21, 2018 (with a 1-minute time resolution) at Dongying site. Concentrations of
these species were measured by an Aerodyne Quadruple Aerosol Chemical Speciation Monitor (ACSM;
Aerodyne Research Inc., USA, Ng et al. (2011)) from November 11 to December 18 in 2018, with a time
resolution of 2 minutes at Gucheng site (Li et al., 2021).
Concentrations of $N_2O_5$ and $ClNO_2$ (with a time resolution of 1 minute) were also measured at
Guangzhou site by a Chemical Ionization Mass Spectrometer (CIMS, THS Instruments Inc., Atlanta,
(Kercher et al., 2009)) from September 25 to November 12 in 2018 (Ye et al., 2021). To have a thorough
evaluation of the representativeness of different parameterizations for $\gamma_{N2O5}$ and $\varphi_{ClNO2}$, observations of
$ClNO_2$ and $N_2O_5$ at six more sites across China from previous studies (see Table S2 and Fig. S2) are also
used in this study. It should be noted that model results sampled at those sites for comparison were
simulated in the same months but different years while ignoring the uncertainties associated with the
interannual variability.
In addition, we also use observed hourly data of $O_3$ and $PM_{2.5}$ published by the China National
Environmental Monitoring Center (CNEMC, http://www.cnemc.cn/sssj/, last access on June 20, 2021)
to evaluate the model's overall performance in China. The network was launched in 2013 as part of the
Clean Air Action Plan and includes ~1500 stations located in 370 cities by 2018 (Fig. S2).

**3 Results and discussion**

**3.1 Improved model performance with updated chlorine emissions and parameterizations for the heterogeneous $N_2O_5$ + Cl chemistry**

Figure 2 shows time series of observed and simulated $Cl^-$ concentrations at the Guangzhou, Dongying, and Gucheng sites. The observations show the lowest $Cl^-$ concentrations at the Guangzhou site ($0.55 \pm 0.52$ µg m$^{-3}$), although the site is the closest to the ocean among all three sites, while the highest concentrations ($4.7 \pm 3.3$ µg m$^{-3}$) are observed at the Gucheng site, away from the sea. Moderate concentrations of $Cl^-$ is observed at the Dongying site, around $1.1 \pm 0.82$ µg m$^{-3}$. The relatively higher concentrations observed inland again suggest the dominance of non-sea salt $Cl^-$ in China, as mentioned before in the Introduction.

The comparison between observations and simulated results from the NoEm case shows a serious underestimate of $Cl^-$ concentrations, with normalized mean bias (NMB) ranging from -96% to -79%, suggesting the missing of significant chlorine sources in addition to sea salt chlorine. In contrast, the Base case with updated chlorine emissions exhibits much higher $Cl^-$ concentrations and can successfully reproduce observations, with average concentrations of $0.77 \pm 0.54$, $0.71 \pm 0.52$, and $4.5 \pm 2.4$ µg m$^{-3}$ (NMB = 39%, -36% and -4.7%) at the Guangzhou, Dongying, and Gucheng sites, respectively. The increase in $Cl^-$ concentrations in the Base case compared with the NoEm case is the most significant at the Gucheng site, by a factor of 24 (from 0.19 to 4.5 µg m$^{-3}$ on average). The slight underestimates at the Dongying site in the Base case could be to some extent explained by the bias in GFED4, which underestimates emissions from agricultural fires due to their small size and short duration as suggested by the study of Zhang et al. (2020). In spite of that, the model with the Base case well reproduces the overall distribution of the observed particulate chloride concentrations in China. The correlation coefficients ($r$) between observed and model results at the three sites also increase from -0.05 – 0.61 in the NoEm case to 0.40 – 0.71 in the Base case. The significant improvement in the model performance again suggests sources other than SSA play a key role in $Cl^-$ concentrations in China.

The comparison between observed and simulated $N_2O_5$ (Fig. 3a) shows that NMB for the NoEm case are

-58%, 150%, 108% and 25% at the Guangzhou, Wangdu, Taizhou and Mount Tai sites, respectively. In
contrast, the corresponding NMB for the Base case are much smaller, -57%, 48%, 91% and 18%,
respectively. The improvement in the Base case is apparent at most sites, implying that additional
chlorine emissions could effectively increase the uptake coefficient of $N_2O_5$ in Yu parameterization. As
shown in Figure S3, although the values of $\gamma_{N2O5}$ between the Base and NoEm cases are similar over the
ocean, the Base case has relatively higher $\gamma_{N2O5}$ over China compared with the NoEm case (0.016 vs.
0.014 on annual mean basis). Little improvement is found at the Guangzhou site (-58% in the NoEm case
vs. -57% in the Base case). Previous studies also found an underestimation of $N_2O_5$ in the Pearl River
Delta, which could be partly explained by the underestimation of the sources (e.g $NO_2$) and/or the
overestimation of the sink of $N_2O_5$ there (Dai et al., 2020; Li et al., 2016).
The $N_2O_5$ results from the McDuffie case, which uses McDuffie parameterization (a default setting in
GEOS-Chem, see Section 2.1.1 and 2.1.3) instead of Yu parameterization are also shown in Fig. 3a. The
NMB for the McDuffie case are -53%, 154%, 143% and 37% at the Guangzhou, Wangdu, Taizhou and
Mount Tai sites, respectively. The comparison between the McDuffie and Base cases indicates that Yu
parameterization can reproduce observed $N_2O_5$ better in China in general, while McDuffie
parameterization tends to overestimate $N_2O_5$ concentrations. The overestimate of $N_2O_5$ in McDuffie
parameterization suggests the potential underestimate in the corresponding $\gamma_{N2O5}$. As shown Figure S3,
the value of $\gamma_{N2O5}$ from the McDuffie case is much smaller than that from the Base case (0.0071 vs. 0.016
averaged over China).
The underestimate in $\gamma_{N2O5}$ from the McDuffie case could to large extent be explained by the suppressive
effect of organic coatings ($\gamma_{coat}$) as discussed above in Section 2.1.1. The magnitude of the organic
suppression is highly dependent on many factors (e.g. organic composition, particle phase state, etc.) and
thus remains poorly quantified (Griffiths et al., 2009; Gross et al., 2009; Thornton et al., 2003). Although
many studies have shown that organic aerosol can suppress the $N_2O_5$ uptake (Anttila et al., 2006; Riemer
et al., 2009), the level of organic suppression may be overpredicted in currently implemented
parameterization attributed to the poorly quantified and/or unknown factors (e.g. Morgan et al. (2015)).
For example, some studies found that ignoring the difference between water-soluble and water-insoluble
organics may lead to an upper limit for the suppressive effect of organic coatings and consequently an
underestimate in the solubility and diffusivity of $N_2O_5$ in organic matter (Chang et al., 2016; Yu et al.,
2020). Although the $\gamma_{coat}$ in McDuffie parameterization is calculated as a function of organic aerosol O:C
ratio and RH (see Eq. 2), which could increase with higher RH and higher O:C ratio, it may still
overpredict the suppressive role of organic coatings in China. On the other hand, the study by Yu et al.
(2020) found that excluding the organic coating best reproduced uptake coefficients observed in China.
In addition, the underestimate in $\gamma_{N2O5}$ in McDuffie parameterization in China could also be to some
extent explained by the lack of the chloride enhancement (also discussed in Section 2.1.1). It is worth
noting that the evaluation here is specific to China and the differences between Yu and McDuffie
parameterizations have not been evaluated elsewhere.
For the comparison of $ClNO_2$ (Fig. 3b), we use the mean nighttime (excluding the data at local time of
10:00 – 16:00) maximum mixing ratio, as suggested by Wang et al. (2019). Observed $ClNO_2$ is high in
Guangzhou (1121 pptv) and Wangdu (~ 990 pptv), followed by Changping (~ 500 pptv) and Beijing (~
430 pptv). The lowest concentrations are obtained at Mount Tai and Mount TaiMoShan (~ 150 and 120
pptv, respectively) due to relatively clean condition at high altitude. The comparison between observed
and simulated $ClNO_2$ at different sites also suggests a better model performance for the Base case with
NMB in the range of -28% – 22%, compared with the NMB of -77% – -31% and -59% – -36% for the
NoEm and McDuffie cases, respectively. The difference in $ClNO_2$ concentrations is mainly associated
with distinct $\varphi_{ClNO2}$ values among different cases. As shown in Figure S4, the value of $\varphi_{ClNO2}$ is
significantly higher in the Base case (0.36 averaged over China) than in the NoEm (0.14) and McDuffie
(0.11) cases. The large difference between the NoEm and Base cases again emphasizes the important role
of non-sea salt chlorine in the formation of $ClNO_2$. The overall underestimates in McDuffie
parameterization on the other hand may suggest that the scaling factor of 0.25 applied to $\varphi_{ClNO2}$ in Eq. 4
is too much for the atmospheric condition in China. More field measurements and model evaluations are
required to come up with a more precise parameterization better representing $\varphi_{ClNO2}$ in China.
Overall, with updated chlorine emissions and Yu parameterization for $\gamma_{N2O5}$ and $\varphi_{ClNO2}$, the Base case
agrees better with both the magnitude and the spatial variation of observed $N_2O_5$ and $ClNO_2$ in China.
The differences in $\gamma_{N2O5}$ and $\varphi_{ClNO2}$ could also affect the ratios of $ClNO_2$ to $HNO_3$. As shown in Figure
S5, the value of $ClNO_2/HNO_3$ is the highest from the Base case (9.8% averaged in China and up to 47%

in the Sichuan Basin on annual mean basis), followed by the McDuffie (4.7% averaged in China and up to 18% in the Sichuan Basin) and NoEm (3.1% averaged in China and up to 12% in coastal regions) cases.

To further elucidate how the model behaves in reproducing the spatial distribution of ozone and $PM_{2.5}$ through the incorporation of the additional chlorine emissions and Yu parameterization for the heterogeneous $N_2O_5$ ＋ Cl chemistry, simulated MDA8 $O_3$ and $PM_{2.5}$ from different cases were compared with observations across China. Figure 4 shows simulated annual mean MDA8 $O_3$ and $PM_{2.5}$ in 2018 in China from the Base case compared with the observations from CNEMC (China National Environmental Monitoring Center, introduced in Section 2.2). The observed annual mean MDA8 $O_3$ and $PM_{2.5}$ are 49 ppbv and 39 μg m$^{-3}$ respectively in 2018 in China. Model results from the Base case could generally reproduce observed spatial and seasonal variations of annual mean MDA8 $O_3$ and $PM_{2.5}$ concentrations, with NMB of -26% and 3.6% and $r$ of 0.83 and 0.81 respectively (Fig. S6).

Table 3 also summarized the model performance on both annual and seasonal scales regarding the simulation of $O_3$ and $PM_{2.5}$ from different cases. For the comparison with observed MDA8 $O_3$, although different simulation cases show a similar range of $r$, the Base case tends to have a slightly smaller bias in general, with NMB of -26% on annual average (-49% – -5.5% on seasonal mean) vs. -28% (-54% – -5.9%) in the NoEm and -27% (-53% – -5.2%) in the McDuffie case. For the comparison with observed $PM_{2.5}$, the NMB bias from the Base case is 3.6% on annual average (-6.3% – 28% on seasonal mean). Compared with the NoEm case, there is some improvement in summer (5.0% vs. 3.9%) and winter (-7.9% vs. -4.3%) but slightly larger bias in autumn (26% vs. 28%). The McDuffie case on the other hand produces slightly higher $PM_{2.5}$ concentrations, with NMB of 5.6%. Regarding the chemical compositions of $PM_{2.5}$ (Table S3), although the model performance varies with sites and species, the Base case demonstrates better agreement with observations compared with the NoEm and McDuffie cases in general.

On the whole, the model performance is better with the additional anthropogenic and biomass burning chlorine emissions combined with Yu parameterization for the heterogeneous $N_2O_5$ ＋ Cl chemistry. Therefore, the following investigation of the impacts of chlorine chemistry on air quality in China as well as their sensitivities to chlorine emissions and parameterizations for the heterogeneous $N_2O_5$ ＋ Cl

chemistry is mainly based on the Base case.

**3.2 Impacts of tropospheric chlorine chemistry on air quality and the role of the heterogeneous**

**$N_2O_5$ + Cl chemistry**

To comprehensively quantify the importance of chlorine chemistry, we conducted a sensitivity case in
which all related tropospheric chlorine chemistry was turned off (NoChem, also listed in Table 2). The
differences between the Base and NoChem cases (Fig. 5 and Fig. S7) could thus represent the impact of
the chlorine chemistry. The comparison shows that chlorine chemistry could increase annual mean
nighttime max $ClNO_2$ surface concentrations by 243 pptv averaged in China (up to 1548 pptv in the
Sichuan Basin). The increase in annual mean Cl atoms is $1.7 \times 10^3$ molec cm$^{-3}$ averaged in China (up to
$7 \times 10^3$ molec cm$^{-3}$ in coastal regions). The increased Cl atoms could react with VOCs (especially alkanes)
producing more peroxy radicals, including organic peroxy radicals ($RO_2$) and hydroperoxyl radicals
($HO_2$). As shown in Figure S7 (a), the chlorine chemistry could increase annual mean $HO_2$ concentrations
by $1.6 \times 10^6$ molec cm$^{-3}$ averaged in China (up to $8.6 \times 10^6$ molec cm$^{-3}$ in the coastal regions). In the
presence of NO, the peroxy radicals recycle OH while oxidize NO to $NO_2$. The subsequent photolysis of
$NO_2$ could further lead to more $O_3$ production and consequently also more OH (Osthoff et al., 2008;
Riedel et al., 2014; Simpson et al., 2015). On the other hand, the recycling of $NO_x$ back into the
atmosphere associated with the photolysis of $ClNO_2$ could also lead to more $O_3$ production. The results
here show a significant increase in surface annual mean OH (Fig. S7 (b)) and MDA8 $O_3$ (Fig. 5c) by 3.8
$\times 10^4$ molec cm$^{-3}$ and 1.1 ppbv respectively averaged in China (up to $1.2 \times 10^5$ molec cm$^{-3}$ and 4.5
ppbv respectively in the Sichuan Basin). In contrast, annual mean $PM_{2.5}$ surface concentrations are
decreased by 0.91 μg m$^{-3}$ averaged in China (up to 7.9 μg m$^{-3}$ in the Sichuan Basin), mainly due to the
decrease of $NO_3^-$ and $NH_4^+$ (up to 6.4 μg m$^{-3}$ and 1.9 μg m$^{-3}$ respectively), although $SO_4^{2-}$ concentrations
are increased slightly by up to 1.2 μg m$^{-3}$ in the Sichuan Basin (Fig. S7).
Both global and regional studies suggested that the heterogeneous $N_2O_5$ + Cl chemistry can enhance $O_3$
production through the production of Cl atoms and the recycling of $NO_x$ (Li et al., 2016; Sarwar et al.,
2014; Wang et al., 2019). Therefore, we further investigate the role that the heterogeneous $N_2O_5$ + Cl
chemistry plays in tropospheric chlorine chemistry through the comparison between the Base and NoHet
(Fig. 6 and Fig. S8) cases. Keep it in mind that the comparison is mainly assessing the impact of $ClNO_2$
production, namely the uptake of $N_2O_5$ on chloride aerosol, not the general role of $N_2O_5$ heterogeneous
chemistry. The comparison illustrates that the heterogeneous $N_2O_5 + Cl$ chemistry could result in a
significant production of $ClNO_2$, reaching $600 - 1400$ pptv for annual mean nighttime max surface
concentrations in the North China Plain, and up to 1546 pptv in the Sichuan Basin. The change in the
surface concentrations of Cl atoms (an annual mean increase of $1 - 4 \times 10^3$ molec $cm^{-3}$ in central and
eastern China) is mainly due to the photolysis of $ClNO_2$ and accounts for 74% of total change in annual
mean Cl atoms due to all tropospheric chlorine chemistry in China, which are consistent with the results
from the previous study by Liu et al. (2017).
In addition to the production of Cl atoms, the $ClNO_2$ formation also affects the partitioning of $NO_y$ from
$HNO_3$ into more reactive forms (e.g., $NO_x$ and $ClNO_2$) through the recycling of $NO_x$, and therefore of
great importance in atmospheric chemistry (Bertram et al., 2013; Li et al., 2016; Wang et al., 2020a). To
analyze the impact of the heterogeneous $N_2O_5 + Cl$ chemistry on $NO_y$ partitioning, Figure S9 shows the
change in the ratios of $NO_x$ to $NO_y$ and $NO_3^-$ to $NO_y$ as the difference between the Base and NoHet cases.
Since $ClNO_2$ could be treated as a reservoir for reactive nitrogen at night, we include $ClNO_2$ as part of
$NO_x$ in the calculation ($NO_x = NO + NO_2 + ClNO_2$ and $NO_y = NO + NO_2 + ClNO_2 + HNO_3 + 2 \times N_2O_5$
$+ NO_3 + HONO + HNO_4 + NO_3^- +$ various organic nitrates). The results show that due to the $ClNO_2$
production, the ratios of $NO_x$ to $NO_y$ increase by 1.8% averaged in China and up to 5.4% in the Sichuan
Basin, Northeast Plain and North China Plain on annual mean basis. Meanwhile, the ratios of $NO_3^-$ to
$NO_y$ decrease by 1.1% averaged in China and up to 5.1% in the Sichuan Basin on annual mean basis.
Consequently, the annual mean MDA8 $O_3$ surface concentrations are increased by $1.5 - 3$ ppbv in central
and eastern China and up to 3.8 ppbv in the Sichuan Basin, accounting for 83% of total change in annual
mean MDA8 $O_3$ due to all tropospheric chlorine chemistry in China. It is interesting to note that while
MDA8 $O_3$ surface concentrations show maxima in summer and minima in winter in general, the influence
of the heterogeneous $N_2O_5 + Cl$ chemistry on $O_3$ concentrations exhibits a different seasonality. The
increase in seasonal mean MDA8 $O_3$ concentrations is the largest in winter (by up to 6.5 ppbv in the
Sichuan Basin) but is less than 1.5 ppbv in summer. This is because of more accumulation of $N_2O_5$ and
$ClNO_2$ in dark conditions in long winter nights (Sarwar et al., 2014).
There also exhibits an obvious decrease in the annual mean surface concentrations of $PM_{2.5}$ attributed to
the heterogeneous $N_2O_5$ + Cl chemistry, ranging from 1.5 to 4.5 μg m$^{-3}$ in central and eastern China
(accounting for 90% of total change in annual mean $PM_{2.5}$ due to all tropospheric chlorine chemistry in
China). The decrease is more significant in autumn and winter in China, with a range of 3.5 – 5.5 μg m$^{-}$
$^{3}$ in central and eastern China and being up to 11 μg m$^{-3}$ in the Sichuan Basin. In contrast, the decrease in
$PM_{2.5}$ is less than 2 μg m$^{-3}$ in summer in China. The change in $PM_{2.5}$ is mainly due to the decrease in
$NO_3^-$ (up to 6.2 μg m$^{-3}$ in the Sichuan Basin on annual average). In addition, $NH_4^+$ is also decreased by
up to 1.8 μg m$^{-3}$ in the Sichuan Basin on annual average, following the pattern of $\Delta NO_3^-$. This is because
$NH_3$ is in excess in most regions in China (Xu et al., 2019) and the formation of $ClNO_2$ via R2 could
hinder the formation of $HNO_3$ and shift the partitioning between $NH_3$ and $NH_4^+$ towards $NH_3$. Unlike the
change in $NO_3^-$ and $NH_4^+$, the heterogeneous $N_2O_5$ + Cl chemistry increases surface $SO_4^{2-}$ concentration
slightly, which could be explained by the enhancements of atmospheric oxidation associated with the
increase in Cl atoms, OH and $O_3$, facilitating the formation of secondary aerosols (Sarwar et al., 2014).
On the other hand, the effect of tropospheric chlorine chemistry without the heterogeneous $N_2O_5$ + Cl
chemistry is much smaller (Fig. S10, the comparison between the NoHet and NoChem cases), leading to
an increase of up to 0.7 ppbv in inland China and a decrease of 0.3 – 0.5 ppbv in coastal regions for
annual mean MDA8 $O_3$ concentrations. The increase is probably associated with Cl atoms from
photolysis of gas-phase chlorine, especially non-sea salt $Cl_2$ in inland China, while the decrease at coastal
regions is mainly due to catalytic production of bromine and iodine radicals originated from sea-salt
aerosols. The comparison demonstrates the dominance of the heterogeneous $N_2O_5$ + Cl chemistry in
total tropospheric chlorine chemistry in China.
**3.3 The effect of heterogeneous $N_2O_5$ + Cl chemistry in response to chlorine emissions**
Since both $\gamma_{N2O5}$ and $\varphi_{ClNO2}$ in Yu parameterization are highly dependent on [Cl$^-$], the effect of the
heterogeneous $N_2O_5$ + Cl chemistry on air quality is thus sensitive to chlorine emissions. Figure 7 shows
the effects of the additional chlorine emissions from anthropogenic and biomass burring sources on
annual mean surface concentrations of different species (Cl$^-$, Cl atom, OH, MDA8 $O_3$, $PM_{2.5}$ and $NO_3^-$)
in China, calculated as the differences between the Base and the NoEm case. With the implementation
of the additional chlorine emissions, the particulate Cl$^-$ concentration increased significantly in inland
China, with the largest increase in the Sichuan Basin (4.5 μg m$^{-3}$) and little change in west China. The
increase is in the range of 1.5 – 3.5 μg m$^{-3}$ in the North China Plain and < 0.5 μg m$^{-3}$ in South China. The
spatial distribution of ΔCl atoms is also consistent with that of the additional chlorine emissions and ΔCl$^-$,
showing the largest increment in the Sichuan Basin (about 4.5 – 5 × 10$^3$ molec cm$^{-3}$). There is also a
moderate increase in Cl atoms in the Northeast Plain and North China Plain, with a range of 1.5 – 4 ×
10$^3$ molec cm$^{-3}$. Only a minor increase of Cl atoms is found in South China (< 1 × 10$^3$ molec cm$^{-3}$).
As discussed earlier in Section 3.2, increased Cl atoms could lead to more HO$_2$ and OH via VOCs
oxidation. Combined with increased NO$_x$ associated with the release of NO$_2$ upon the photolysis of
ClNO$_2$, further increases in both O$_3$ and OH could also be expected. The increase in OH is around 2 – 9
× 10$^4$ molec cm$^{-3}$ in central and eastern China on annual mean basis. The increase in MDA8 O$_3$ surface
concentrations ranges from 0.5 to 3 ppbv in central and eastern China and reaches up to 3.5 ppbv in the
Sichuan Basin on annual average. The impacts of chlorine sources on O$_3$ formation also vary with seasons.
Although O$_3$ pollution is generally severe in summer, the change in MDA8 O$_3$ due to the additional
chlorine sources is relatively minor, with maxima of 0.7 ppbv in the Sichuan Basin and < 0.5 ppbv in
most other regions averaged in summer. In contrast, the increase is most obvious in winter, with maxima
of 5.2 ppbv in the Sichuan Basin on seasonal average.
The effects of the additional chlorine emissions on surface PM$_{2.5}$ concentrations are complicated. The
North China Plain shows the largest increase (3 – 4.5 μg m$^{-3}$ on annual average), mainly due to the
increase in Cl$^-$, which could also promote the heterogeneous N$_2$O$_5$ + Cl chemistry and lead to more NO$_3^-$
production (Chen et al., 2021a). In contrast, the Sichuan Basin exhibits both an increase (by up to 4.2 μg
m$^{-3}$) and a decease (by up to 3.7 μg m$^{-3}$). The decrease of PM$_{2.5}$ in the Sichuan Basin is mainly due to the
large decrease of NO$_3^-$ there. In the Sichuan Basin, nitrate formation is dominated by the heterogeneous
hydrolysis of N$_2$O$_5$ (Tian et al., 2019) while the additional Cl$^-$ could hinder the path of N$_2$O$_5$ hydrolysis
due to the competition with the path of ClNO$_2$ formation. Consequently, the additional chlorine emissions
result in a decrease of NO$_3^-$ up to 5.6 μg m$^{-3}$ in the Sichuan Basin on annual average.
In addition, NH$_4^+$ concentrations could also be affected through the reaction of R5:
$HCl (g) + NH_3 (g) \rightarrow NH_4^+ + Cl^-$                       (R5)
In the Northeast Plain and North China Plain where anthropogenic and biomass burning emissions of
HCl are high, the annual mean $NH_4^+$ surface concentrations are increased by $0.5 - 1.5$ $\mu g$ $m^{-3}$ (Fig. S11
(a)). $NH_4^+$ concentrations are also affected by the gas-particle partitioning equilibrium, and decease as
the pH value gets higher (or increase with $H^+$ concentrations). Therefore, the competition between the
heterogeneous $N_2O_5$ + Cl chemistry and $N_2O_5$ hydrolysis could also affect the formation of $NH_4^+$. In
other words, increased $Cl^-$ concentrations could results in less $H^+$ and thus less $NH_4^+$. Consequently, there
also exits some decrease in $NH_4^+$ concentrations in the Sichuan Basin associated with the large decrease
in $NO_3^-$ concentrations. In contrast, little change is found for surface $SO_4^{2-}$ concentrations, less than 0.5
$\mu g$ $m^{-3}$ in most regions of China (Fig. S11 (b)).
It is worth mentioning that the effects of the additional chlorine emissions work mainly through the
heterogeneous $N_2O_5$ + Cl chemistry. Without this heterogeneous chemistry, the increase of chlorine
emissions shows only a minor change in Cl atoms ($< 10^3$ molec $cm^{-3}$ in China, estimated as the difference
between the NoHet and NoEmHet cases in Fig. S12). The impact of chlorine emissions on $O_3$
concentrations also weakens when the heterogeneous $N_2O_5$ + Cl chemistry is turned off, with an increase
of $0.5 - 1$ ppbv in MDA8 $O_3$ on annual average (vs. $0.5 - 3$ ppbv mentioned above).
On the other hand, the impacts of heterogeneous $N_2O_5$ + Cl chemistry on air quality in inland China
would be seriously underestimated if the additional anthropogenic and biomass burning chlorine sources
are ignored. If only sea salt chlorine emission is included in the simulation, the increase of $ClNO_2$ surface
concentrations resulted from heterogeneous $N_2O_5$ + Cl chemistry only occurs in coastal regions due to
the heterogeneous uptake of $N_2O_5$ on sea salt chloride aerosols (by up to 260 pptv on annual average,
indicated by the difference between the NoEm and NoEmHet cases, Fig. S13). Consequently, the increase
in Cl atoms and MDA8 $O_3$ surface concentrations is found mainly in coastal regions. For instance, annual
mean MDA8 $O_3$ concentrations are increased by up to 2 ppbv in coastal regions, but by less than 0.5
ppbv in inland China. In other words, the dominance of the heterogeneous $N_2O_5$ + Cl chemistry in the
impact of chlorine chemistry on air quality in China is to large extent driven by the additional chlorine
emissions.

**3.4 The effect of heterogeneous $N_2O_5$ + Cl chemistry in response to parameterizations for $\gamma_{N2O5}$ and $\varphi_{ClNO2}$**

It should be noted that the impact of the heterogeneous $N_2O_5$ + Cl chemistry on air quality not only depends on the amount of chlorine emissions, but is also sensitive to the parameterizations for $\gamma_{N2O5}$ and $\varphi_{ClNO2}$. As discussed earlier (Fig. 3a), there exists a large difference in simulated $N_2O_5$ between the Base and NoEm cases at the Wangdu site, implying the sensitivity of $\gamma_{N2O5}$ to chlorine emissions in Yu parameterization and thus the importance of non-sea salt chlorine emissions in China. This is consistent with the dependence on chloride in Yu parameterization, which is included to better reproduce $\gamma_{N2O5}$ observations in China (Yu et al., 2020). The comparison between the Base and NoHet cases ($\gamma_{N2O5}$ = 0.016 and 0.014, respectively) also suggests that the heterogeneous uptake of $N_2O_5$ on chloride-containing aerosol surfaces in Yu parameterization is an important loss pathway of $N_2O_5$ and should not be ignored.

Unlike Yu parameterization, $N_2O_5$ concentrations have little dependence on chlorine emissions in McDuffie parameterization (Fig. 3a). This insensitivity to chlorine emissions could be expected from Eq. 2 where the dependence on aerosol chloride is not included so as to better reproduce wintertime reactive nitrogen observations in the eastern U.S. The little dependence of $\gamma_{N2O5}$ on concentrations of Cl$^-$ together with the lower value of $\varphi_{ClNO2}$ make the results from the McDuffie case less sensitive to chlorine emissions, producing less $ClNO_2$ and Cl atoms compared with the Base case (with Yu parameterization) although with the same emission. Consequently, the McDuffie case produces less $O_3$, with annual mean surface concentrations of MDA8 $O_3$ lower by 0.47 ppbv averaged in China (by up to 2 ppbv in the Sichuan Basin), but results in more $PM_{2.5}$ (0.63 μg m$^{-3}$ averaged in China and up to 4.7 μg m$^{-3}$ in the Sichuan Basin on annual mean basis mainly due to changes in $NO_3^-$) (Fig. 8). In other words, compared to the Base case with Yu parameterization, the impacts of chlorine emissions on annual MDA8 $O_3$ and $PM_{2.5}$ in the McDuffie case has been decreased by 48% and 27% respectively averaged in China. Therefore, even with the same amounts of chlorine emissions, the impacts of the heterogeneous $N_2O_5$ + Cl chemistry on air quality vary significantly with different parameterizations.

## 4 Conclusions

Considering the importance of chlorine chemistry in modulating the $O_3$ and $PM_{2.5}$ as well as the previously ignored chlorine emission from anthropogenic and biomass burning, we updated the GOES-Chem model in this study with comprehensive chlorine emissions and a new parameterization based on the study of Yu et al. (2020) for the heterogeneous $N_2O_5$ + Cl chemistry, followed by the extensive evaluation of model performance. Through the utilization of a large number of observational datasets, we found a substantial improvement has been achieved by the additional chlorine emissions, with NMB decreasing from -96% – -79% to -36% – 39% for $Cl^-$ simulation. The comparison with observed $N_2O_5$ and $ClNO_2$ also indicates better model performance with Yu parameterization while $\gamma_{N2O5}$ and $\varphi_{ClNO2}$ are underestimated in McDuffie parameterization (a default setting in GEOS-Chem), resulting in larger model bias. The simulation of $O_3$ and $PM_{2.5}$ also agrees better with observations in general in the Base case (with the additional chlorine emissions and Yu parameterization) than the others.

Total tropospheric chlorine chemistry could increase Cl atoms by up to $7 \times 10^3$ molec $cm^{-3}$, and leads to an increase of up to 4.5 ppbv in MDA8 $O_3$ but a decrease of up to 7.9 μg $m^{-3}$ in $PM_{2.5}$ concentrations on an annual mean basis in China. The decrease in $PM_{2.5}$ is mainly associated with the decrease in $NO_3^-$ and $NH_4^+$, by up to 6.4 and 1.9 μg $m^{-3}$, respectively. The results also indicate that the heterogeneous $N_2O_5$ + Cl chemistry dominate the impact of chlorine chemistry, accounting for 83% and 90% of total change in $O_3$ and $PM_{2.5}$ concentrations. In other words, the chlorine chemistry without the heterogeneous $N_2O_5$ + Cl chemistry has a minor effect on annual mean MDA8 $O_3$ (less than 0.7 ppbv) and $PM_{2.5}$ (less than 1.5 μg $m^{-3}$) concentrations in China. This mechanism is particularly useful in elucidating the commonly seen $O_3$ underestimations relative to observations (e.g. (Ma et al. (2019))).

The effect of the heterogeneous $N_2O_5$ + Cl chemistry is sensitive to chlorine emissions. With the additional anthropogenic and biomass burning sources, simulated $PM_{2.5}$ concentrations are increased by up to 4.5 μg $m^{-3}$ in the North China Plain but decreased by up to 3.7 μg $m^{-3}$ in the Sichuan Basin on an annual basis. The latter is mainly driven by the decrease of $NO_3^-$ due to the competition between the formation of $ClNO_2$ and $HNO_3$ upon the uptake of $N_2O_5$ on aerosol surfaces. The additional emissions also increase Cl atoms and OH in China associated with the photolysis of $ClNO_2$, consequently leading

605 to an increase of annual mean MDA8 $O_3$ concentrations by up to 3.5 ppbv. In contrast, the significance

606 of the heterogeneous $N_2O_5$ + Cl chemistry especially over inland China would be severely

607 underestimated if only sea salt chlorine is considered, with only a slight increase in MDA8 $O_3$ (< 0.5

608 ppbv) and a minor decrease in $PM_{2.5}$ (< 1.5 μg m$^{-3}$) in inland China.

609 Moreover, we found the importance of chlorine chemistry not only depends on the amount of emissions,

610 but is also sensitive to the parameterizations for the heterogeneous $N_2O_5$ + Cl chemistry. Although with

611 the same emission, the effects on MDA8 $O_3$ and $PM_{2.5}$ in China from the McDuffie case are lower

612 compared to the results with Yu parameterization: differing by 48% and 27% in the annual average,

613 respectively.

**Acknowledgements**

615 This study is supported by the National key R&D Program of China (2018YFC0213901), the National

616 Natural Science Foundation of China (41907182, 41877303, 91644218, 41877302, 41875156), the

617 Fundamental Research Funds for the Central Universities (21621105), Guangdong Natural Science

618 Funds for Distinguished Young Scholar (grant No. 2018B030306037), the Guangdong Innovative and

619 Entrepreneurial Research Team Program (Research team on atmospheric environmental roles and

620 effects of carbonaceous species: 2016ZT06N263), and Special Fund Project for Science and

621 Technology Innovation Strategy of Guangdong Province (2019B121205004).

**Competing interests.**

623 The authors declare that they have no conflict of interest.

**Data and Code availability**

625 The data used in this study is available upon request from Qiaoqiao Wang (qwang@jnu.edu.cn). The

626 revised codes for different simulations could be downloaded via

627 https://zenodo.org/record/5957287#.YfyNMppBxPZ

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

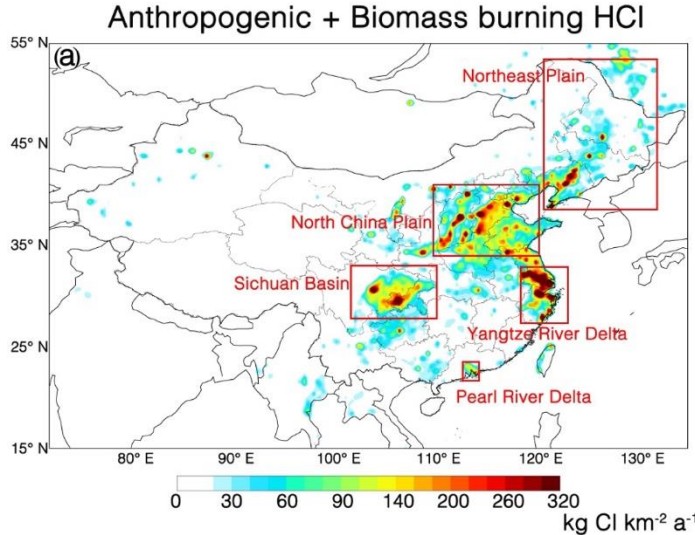

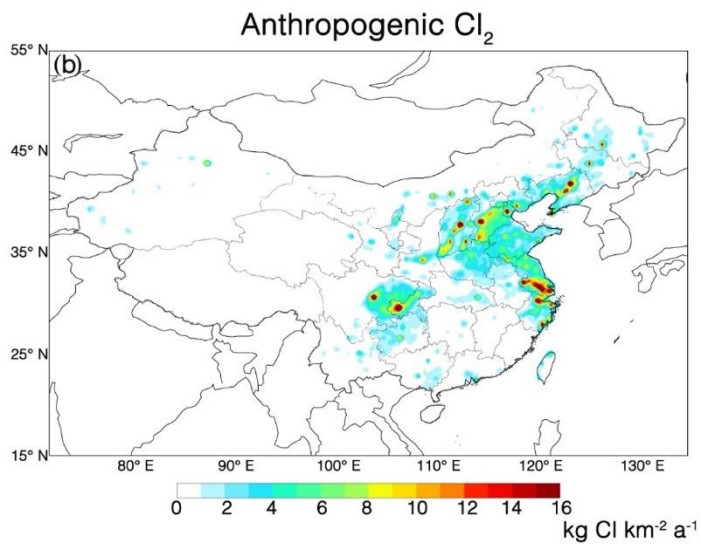

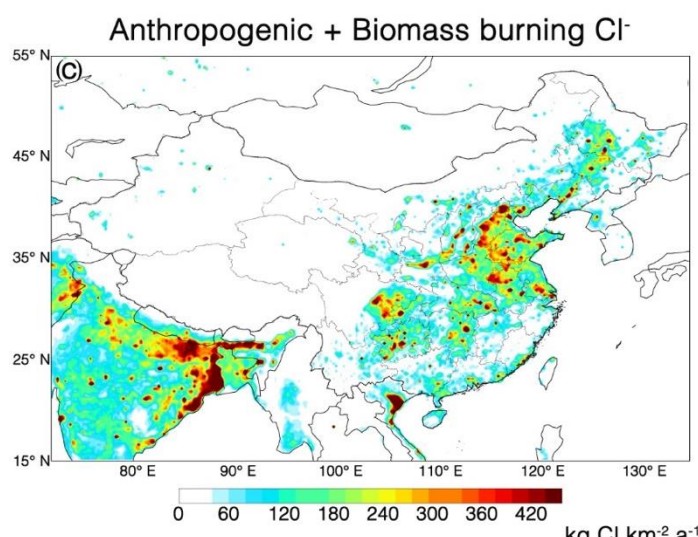

**Figure 1. Annual emissions of (a) HCl, (b) Cl₂, (c) non-sea salt Cl⁻. Locations of the Northeast Plain, North**

**China Plain, Yangtze River Delta, Pearl River Delta and Sichuan Basin are highlighted by red boxes in (a).**

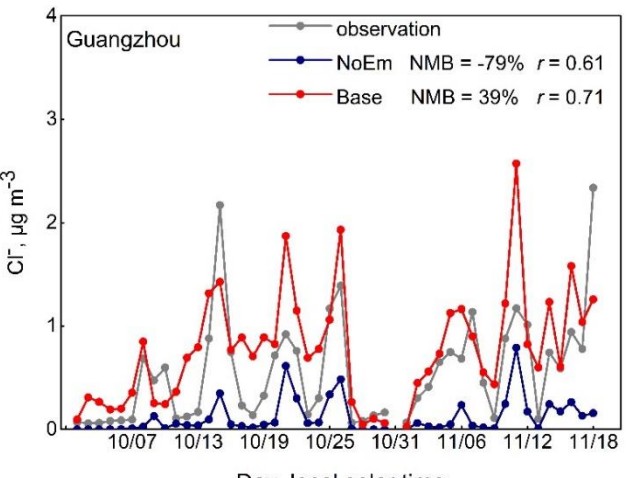

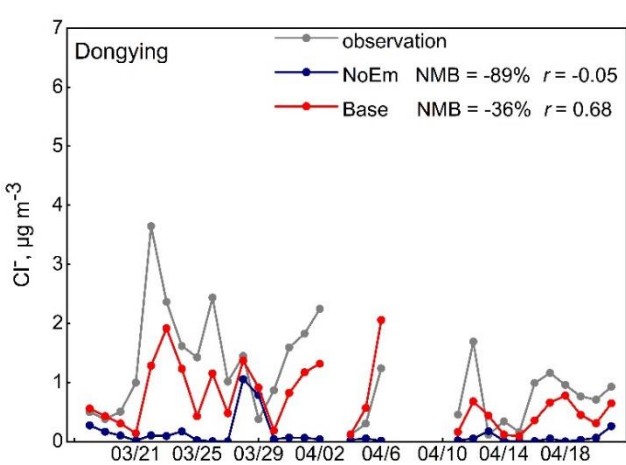

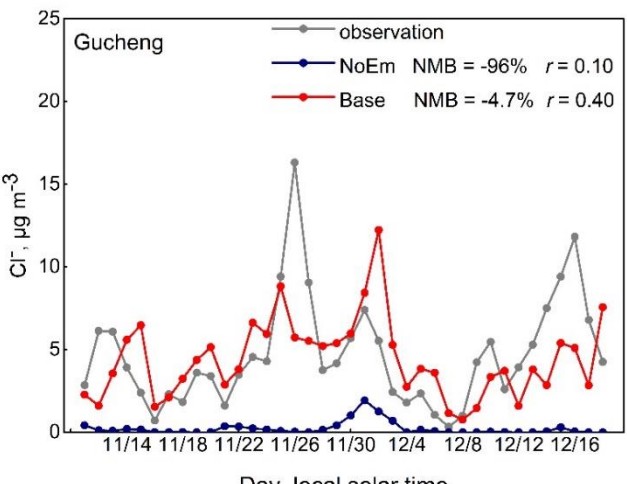


**Figure 2. Time series of simulated and observed particulate Cl⁻ concentrations at the Guangzhou, Dongying**
**and Gucheng sites.**

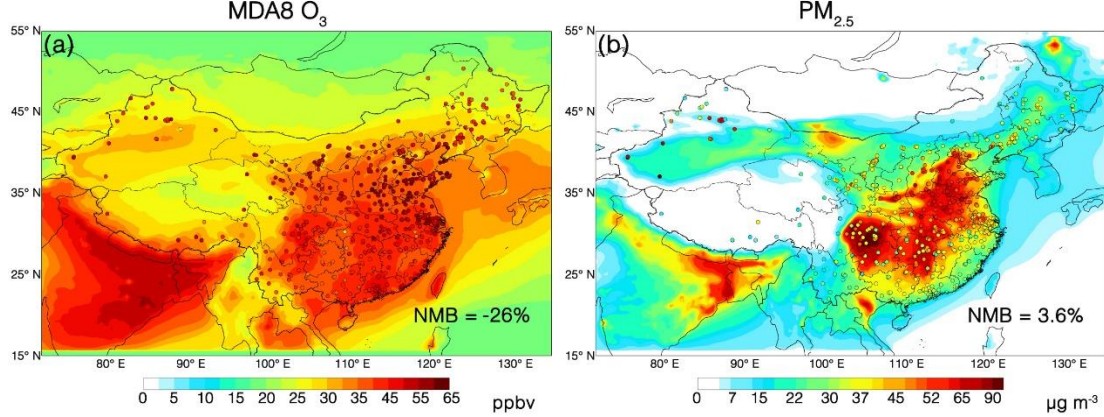


**Figure 3. Comparison of observed and simulated (a) averaged $N_2O_5$ concentrations and (b) mean nighttime**
**maximum mixing ratio of $ClNO_2$ concentrations at different sites. The simulation definitions are provided in**
**Table 2. GZ: Guangzhou; WD: Wangdu; TZ: Taizhou; Tai: Mount Tai; CP: Changping; BJ: Beijing; TMS:**
**Mount TaiMoShan**


**Figure 4. Annual mean surface concentrations of (a) MDA8 $O_3$ and (b) PM2.5 over China in 2018. GEOS-**
**Chem model values from the Base case are shown as contours. Observations from China National**
**Environmental Monitoring Center (CNEMC) are shown as circles.**

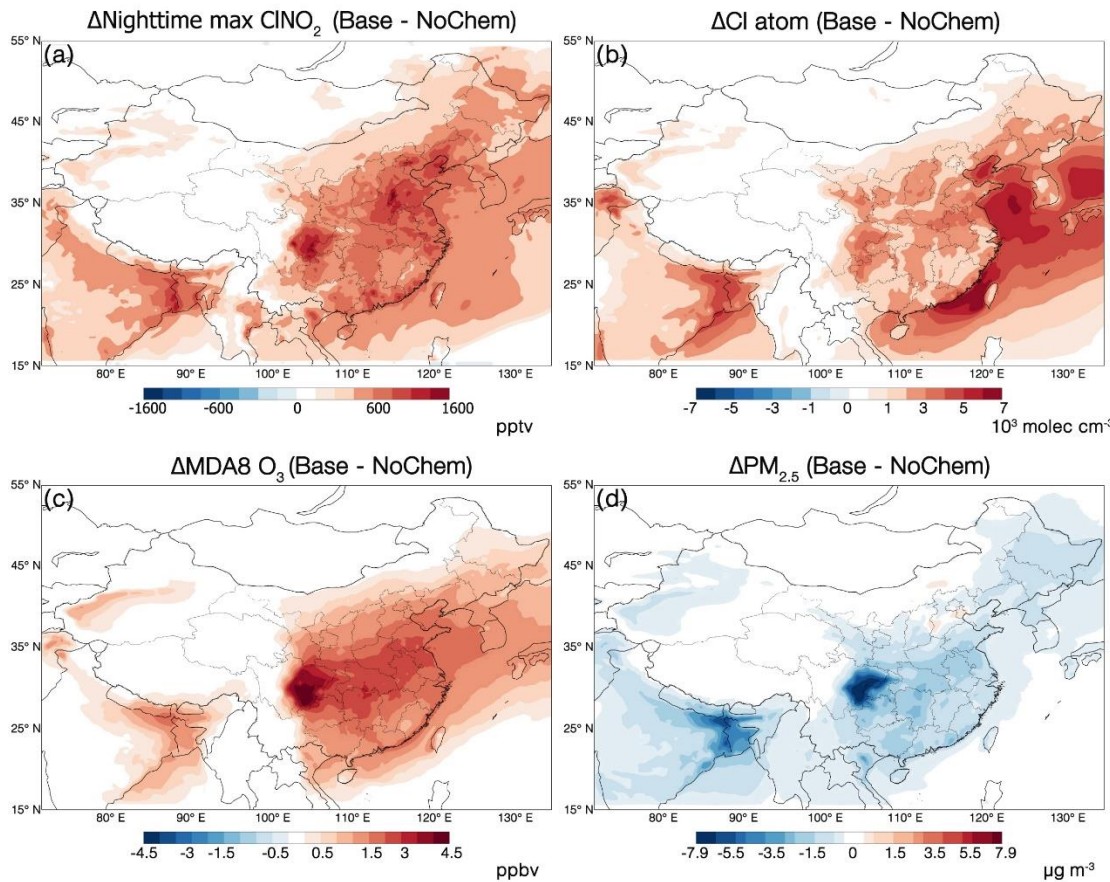

Figure 5. Effects of chlorine chemistry on annual mean surface concentrations of (a) nighttime max $ClNO_2$, (b) Cl atom, (c) MDA8 $O_3$ and (d) $PM_{2.5}$ in China, estimated as the differences between the Base and NoChem cases.

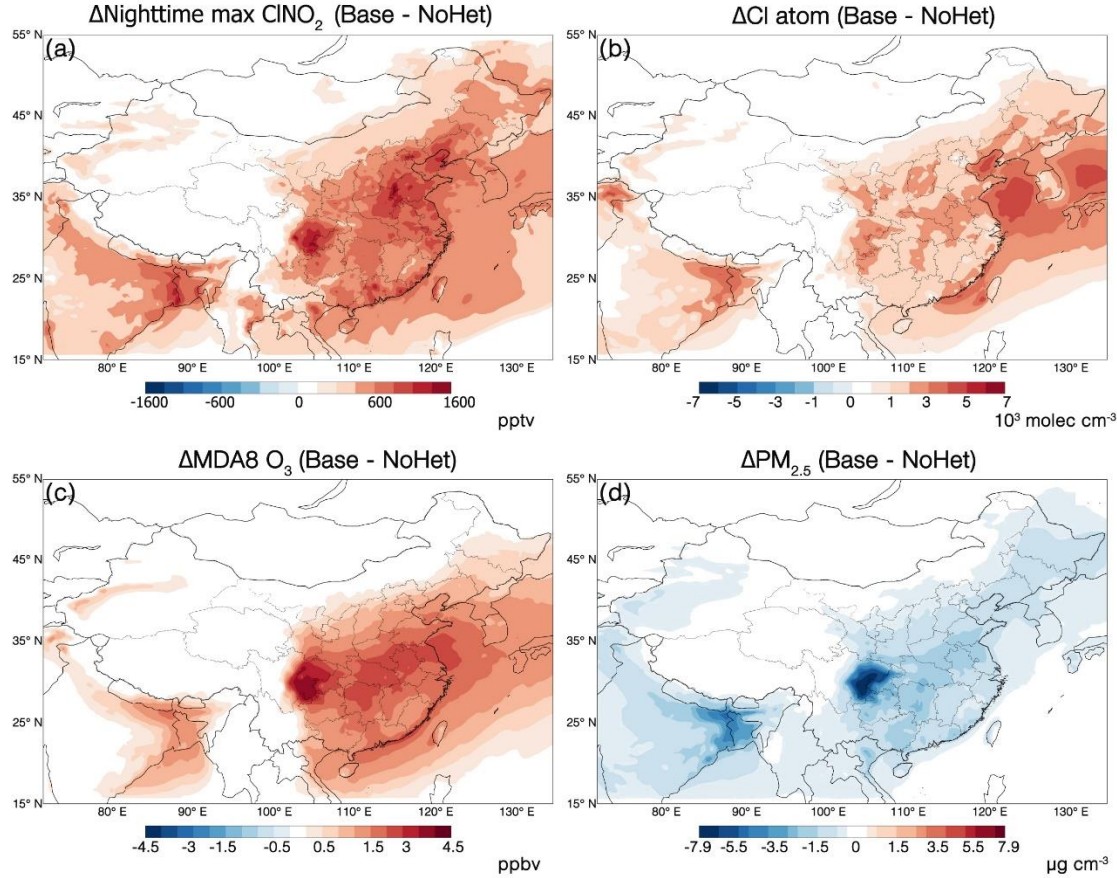

Figure 6. Effects of the heterogeneous $N_2O_5$ + Cl chemistry on annual mean surface concentrations of (a) nighttime max $ClNO_2$, (b) Cl atom, (c) MDA8 $O_3$ and (d) $PM_{2.5}$ in China, estimated as the differences between the Base and NoHet cases.

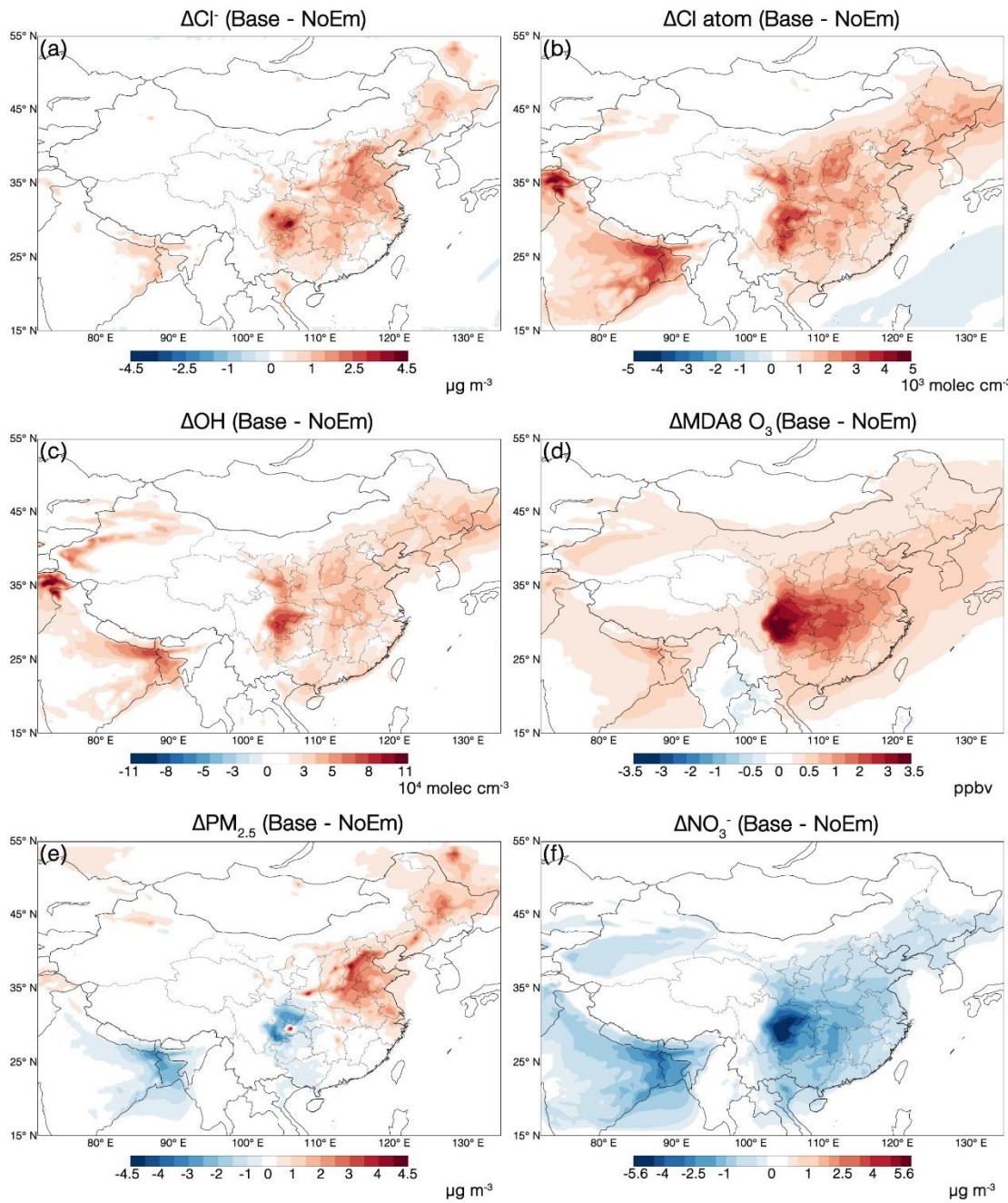

**Figure 7. Effects of anthropogenic and biomass burning chlorine emissions on annual mean surface**
**concentrations of (a) Cl⁻, (b) Cl atom, (c) OH, (d) MDA8 O₃, (e) PM₂.₅ and (f) NO₃⁻ in China, estimated as the**
**differences between the Base and NoEm cases.**

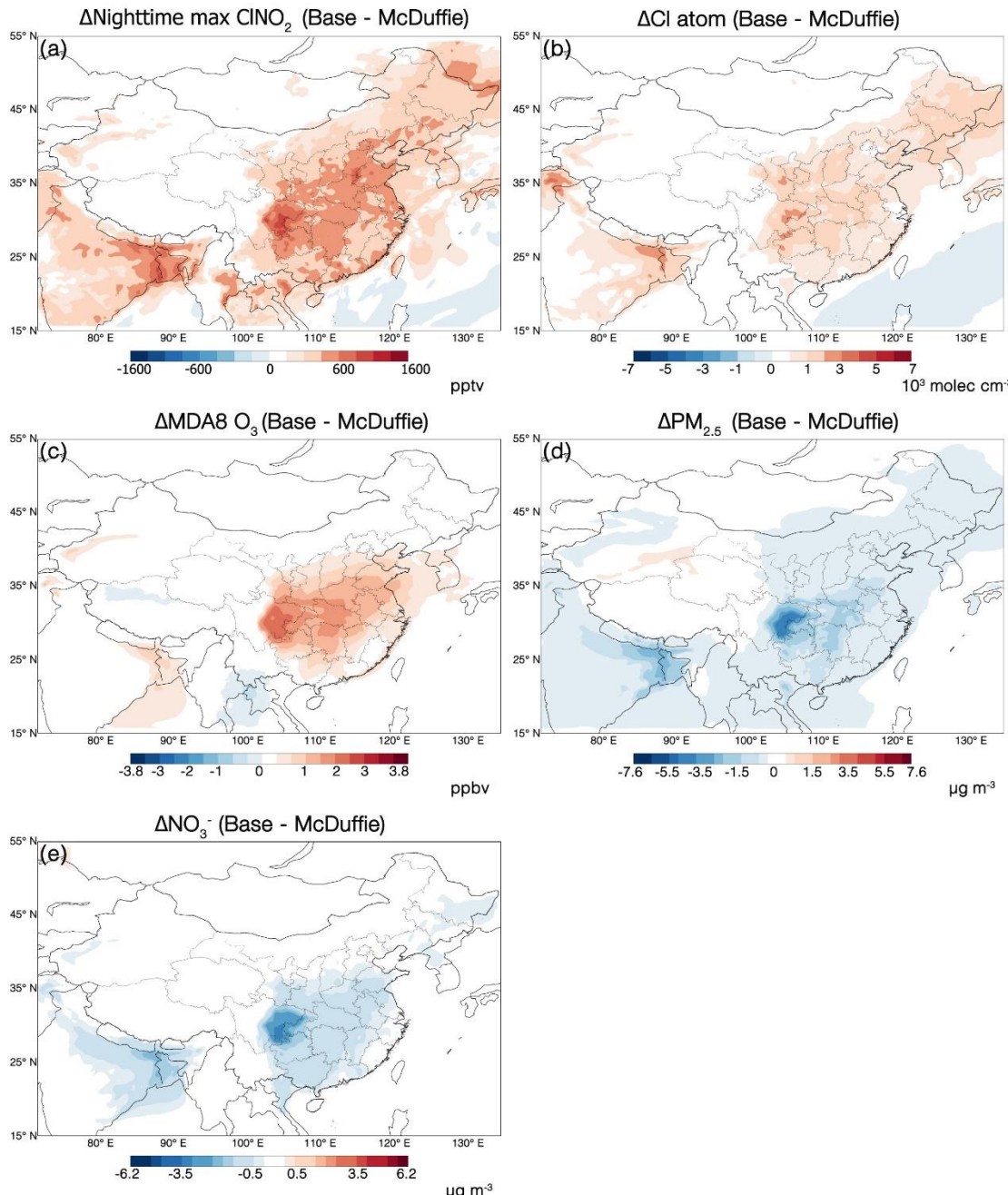

**Figure 8. Effects of different parameterizations on annual mean surface concentrations of (a) nighttime max ClNO₂, (b) Cl atom, (c) MDA8 O₃, (d) PM₂.₅, and (e) NO₃⁻ in China, estimated as the differences between the Base and McDuffie cases.**

**Table 1. Chlorine emissions in China in the model.**

| Sources | By default (Gg Cl a$^{-1}$) | Updated in this study (Gg Cl a$^{-1}$) |
|---|---|---|
| Sea salt Cl$^-$ | $6.5\times10^4$ | $6.5\times10^4$ |
| Anthropogenic HCl | 0 | 218 |
| Biomass burning HCl | 0 | 30 |
| Anthropogenic Cl$_2$ | 0 | 8.9 |
| Anthropogenic Cl$^-$ | 0 | 379 |
| Biomass burning Cl$^-$ | 0 | 120 |
| CH$_3$Cl[a] | 3.8 | 3.8 |
| CH$_2$Cl$_2$[a] | 2.4 | 2.4 |
| CHCl$_3$[a] | 0.70 | 0.70 |

[a]: Sources are shown in terms of the chemical release (e.g. +Cl, +OH, +hv)

**Table 2. Model setup of all simulation cases**

| Cases | N$_2$O$_5$ uptake ($\gamma_{N2O5}$) | ClNO$_2$ production ($\varphi_{ClNO2}$) | Other tropospheric chlorine chemistry | Anthropogenic and biomass burning inorganic chlorine emissions |
|---|---|---|---|---|
| Base | Yu et al. (2020) | Yu et al. (2020) | Full | Yes |
| McDuffie | McDuffie et al. (2018a, 2018b) | McDuffie et al. (2018a, 2018b) | Full | Yes |
| NoEm | Yu et al. (2020) | Yu et al. (2020) | Full | None |
| NoHet | Yu et al. (2020) but with [Cl$^-$] = 0 | None | Full | Yes |
| NoChem | Yu et al. (2020) but with [Cl$^-$] = 0 | None | None | Yes |
| NoEmHet | Yu et al. (2020) but with [Cl$^-$] = 0 | None | Full | None |
| NoAll | Yu et al. (2020) but with [Cl$^-$] = 0 | None | None | None |


**Table 3. Normalized mean bias (NMB) and correlation coefficients (*r*) between observed and simulated**
**MDA8 O$_3$ and PM$_{2.5}$ concentrations during 2018 in China**

| Species | Time | Base NMB | Base *r* | McDuffie NMB | McDuffie *r* | NoEm NMB | NoEm *r* |
|---|---|---|---|---|---|---|---|
| MDA8 O$_3$ | Annual | -26% | 0.83 | -27% | 0.83 | -28% | 0.82 |
| | MAM[a] | -35% | 0.87 | -36% | 0.87 | -36% | 0.87 |
| | JJA[b] | -5.5% | 0.50 | -5.2% | 0.48 | -5.9% | 0.48 |
| | SON[c] | -24% | 0.79 | -26% | 0.78 | -28% | 0.76 |
| | DJF[d] | -49% | 0.81 | -53% | 0.80 | -54% | 0.80 |
| PM$_{2.5}$ | Annual | 3.6% | 0.81 | 5.6% | 0.81 | 2.3% | 0.80 |
| | MAM | -6.3% | 0.52 | -4.9% | 0.53 | -6.2% | 0.52 |
| | JJA | 3.9% | 0.70 | 4.6% | 0.70 | 5.0% | 0.70 |
| | SON | 28% | 0.79 | 32% | 0.80 | 26% | 0.79 |
| | DJF | -4.3% | 0.82 | -2.6% | 0.82 | -7.9% | 0.82 |

[a]: March, April, and May (Spring)
[b]: June, July, and August (Summer)
[c]: September, October, and November (Autumn)
[d]: December, January, and February (Winter)