# Peer review of "The impact of chlorine chemistry combined with"

_Atmospheric Chemistry and Physics, 2021_

## Referee Comment (RC1)

**Review of Yang, X., et al., The impact of chlorine chemistry combined with heterogeneous N$_2$O$_5$ reactions on air quality in China.**

**General Comments**

In this manuscript, the authors present an analysis that uses simulations with the GEOS-Chem model to assess the sensitivity of air quality (e.g., PM$_{2.5}$ and O$_3$) in China to changes in tropospheric chlorine chemistry, chlorine emissions, and the parameterization of heterogeneous N$_2$O$_5$/ClNO2 chemistry. This study also provides an important evaluation of the inclusion of chlorine emissions and updated parameterizations for heterogenous N$_2$O$_5$/ClNO2 chemistry, showing that both updates better reproduce surface observations of N$_2$O$_5$, ClNO2, and particulate chloride as compared to the default GEOS-Chem model performance. This is a well-designed study and well-written manuscript, though I have specific comments and suggestions below to help deepen the discussion and improve the clarity of the text. For example, there are a few cases where it would help if the authors could be more specific about the processes they are referring to when they say 'N$_2$O$_5$-ClNO2 chemistry'. A more detailed discussion on the differences and similarities in the heterogenous parameterizations would also help the reader better interpret the simulation results (see more comments below).

Overall, this study is an important contribution to the field as it highlights that simulations of absolute air pollutant concentrations are sensitive to both chloride emissions and the parameterization of heterogeneous N$_2$O$_5$ and ClNO2 chemistry. It also shows that the model's sensitivity to emissions depends on these heterogenous parameterizations. Not only do these results help improve our understanding of the importance of these processes, but they also show that air quality impacts from anthropogenic chlorine sources should be considered in air quality improvement strategies, and that our ability to accurately predict the magnitude of the reduction benefits will depend on the modeled representation of heterogeneous N$_2$O$_5$ chemistry. I recommend that this study be accepted for publication after the specific comments below are addressed.

**General Comments:**

1. This comment applies to all sections of the manuscript. In cases where the authors simply state that one parameterization performs better than another, it would be helpful if the authors could provide additional discussions on *why* one might be outperforming the other in terms of the processes controlling $N_2O_5$ uptake and ClNO2 yield. For example, the Yu and McDuffie parameterizations for $N_2O_5$ uptake are actually based on the same general parameterization (see specific comment 4), but the differences are that the Yu parameterization does not consider added suppression from organic aerosol (included in McDuffie), includes an uptake enhancement from particulate chloride (not included in McDuffie), and uses different rate coefficient ratios than McDuffie. Therefore, since this study shows that the Yu parameterization is able to better reproduce available surface observations, the results here suggest that in China, it may be important to consider particulate chloride in the uptake of $N_2O_5$ and that organics may not play as important of a suppressive role as shown in previous studies. Including these details helps explain to the reader why one parameterization may be outperforming another and provides insight into specific modeled processes that could be improved. Many of my specific comments below are related to this general comment.

2. Throughout the text, the authors refer to the simulations with the Yu parameterization as simulations with 'updated $N_2O_5$-ClNO2 chemistry'. This is slightly confusing terminology for the reader as it is not actually the chemical reactions of $N_2O_5$ and ClNO2 that are changing between the simulations. Rather, it is the parameterizations of gamma $N_2O_5$ and phi ClNO2 that are changing. I would suggest editing the terminology throughout the paper to reflect this difference. E.g., Replace '$N_2O_5$-ClNO2 chemistry' with something along the lines of 'updated parameterizations for heterogeneous $N_2O_5$ and ClNO2 chemistry'.

3. The authors may also want to consider including maps or averages of model-calculated $N_2O_5$ uptake coefficients and ClNO2 yields. Reporting these values could help future studies compare the results from each parameterization with available field-derived values. The model diagnostics should be able to provide these values. This could be included in the supplement.

**Specific Comments**

1. Line 80-83 – These two sentences are technically correct, but the authors should adjust them to specifically note the possible role of organics (not considered in the (Bertram and Thornton, 2009) study). It is the presence of organic aerosol species that is thought to reduce the uptake of $N_2O_5$ relative to that predicted by the Bertram and Thornton parameterization. It is not just the mixing state that is a possible difference, but the presence of hydrophobic organic aerosol that lead to a potentially complex aerosol mixing state.

2. Line 87-89 – To strengthen this last point, the authors could mention that some of these previous field-based parameterizations were derived from observations in locations with conditions may not be applicable to the highly polluted regions in China. This makes it important to evaluate these parameterizations under different conditions. The authors should also clarify that in this study, a full evaluation of this chemistry is being conducted for China specifically. As written, 'full evaluation' makes it sound as if this study will be conducting a global analysis.

3. Line 93 – My comment is on the statement 'The importance of anthropogenic chlorine emissions, which were ignored in most studies…". While these emissions are not commonly included in modeling studies, there is a clear example in (Wang et al., 2019) where they did not 'ignore' chlorine emissions but rather found that the addition of anthropogenic chlorine emissions in GEOS-Chem resulted in overestimates of HCl observations in the U.S. This is an important study to cite in this section. The authors could still note that anthropogenic emissions in China may be relatively more important in China than in the U.S., which is why it is important to study their impacts here.

4. Lines 131 - 154 – Per my general comment above, it would be helpful in this section for the authors to provide a more detailed narrative on 1) the differences between the various parameterizations, 2) the processes that they are trying to represent, and 3) a brief summary of how each parameterization was derived.

    For example, for (1 and 2), the authors should include in Eq. 1 the detailed equations for both gamma_coat and gamma_core (found in (McDuffie et al., 2018b)). These details are important to include here because the functional form of the Yu parameterization is actually the same as that of gamma_core from the McDuffie parameterization. By showing the full equations, the readers can better understand that the only differences between the Yu and McDuffie uptake parameterizations are that Yu parameterization includes an enhancement from aerosol chloride (not included in McDuffie), does not consider the added resistance from an organic coating (included in McDuffie) and uses different coefficients (reaction rate constant ratios) than McDuffie in the gamma_core equation.

    For (3), to better understand why the parameterizations are different, it would be useful to briefly explain that the form of the McDuffie parameterization is derived from parameterizations proposed in multiple laboratory studies ((Bertram and Thornton, 2009), Riemer et al., 2009, and (Anttila et al., 2006)) to account for the uptake dependence on

aerosol water and nitrate concentrations and added resistance from an organic aerosol coating. The coefficients in this parameterization were then derived by fitting a chemical box model to aircraft observations of $N_2O_5$, ClNO2, $O_3$, and NOx during the winter over the eastern U.S. In contrast, for the Yu parameterization, it would be important to note that the form of this parameterization is from Bertram and Thornton to account for the dependence on aerosol water, nitrate, and chloride concentrations, with coefficients derived from uptake coefficients directly measured on ambient aerosol in two rural sites in China. It would also be important to note that while previous studies have found that organics can suppress $N_2O_5$ uptake, (Yu et al., 2020) found that including the chloride enhancement and excluding the organic coating best reproduced the observed $N_2O_5$ uptake coefficients in their study.

Similarly for Eq 2., replace k2/k3 with 1/450 to make this form consistent with Eq. 4. Also add a note that both forms are from Bertram and Thornton, again with coefficients and scaling factors (in the case of McDuffie) derived from fits to observations over the eastern U.S. (McDuffie) and rural locations in China (Yu).

5. Line 183 – It would be helpful context for the reader to specify which sectors contribute to emissions of HCl, Cl2, and particulate chloride in the inventories.

6. Line 221 – The authors should clarify that the NoHet case not only sets phi ClNO2 to zero, but also removes the enhancement of $N_2O_5$ uptake from aerosol chloride. So this simulation actually tests the model sensitivities to a smaller gamma $N_2O_5$ and zero ClNO2 production.

7. Line 278 – The comparison here is between the Base and NoEm cases. Therefore, it seems that the results imply that additional chlorine emissions could increase the uptake coefficient due to increased aerosol chloride. This comparison does not directly evaluate the Yu parameterization as implied, since the Yu parameterization is included in both Base and NoEm simulations. This sentence should be updated accordingly.

8. Line 290 – There are many other studies (see section 4.2.6 in (McDuffie et al., 2018b)) that have shown that organics can suppress uptake, which should also be referenced here (the authors can still note that Yu found that excluding the organic coating best reproduced uptake coefficients observed in China). In addition, (Morgan et al., 2015) actually state that "An additional suppression of the parameterised ($N_2O_5$) uptake is likely required to fully capture the variation in N2O5 uptake, which could be achieved via the known suppression by organic aerosol. However, existing parameterisations representing the suppression by organic aerosol were unable to fully represent the variation in N2O5 uptake." Therefore, the sentence should be amended to clarify that organic suppression may be important to consider in the estimate of $N_2O_5$ uptake, but that the currently implemented parameterization may overpredict the level of suppression.

9. Line 293 – 297 – The McDuffie parameterization is slightly more sophisticated than indicated in this sentence. For example, in the McDuffie parameterization, the gamma_coat value is actually calculated as a function of organic aerosol O:C ratio and RH. These factors

are meant to account for conditions where higher relative humidity and higher O:C ratio may represent less likely liquid-liquid aerosol phase separation, a partially coated aerosol, or thinner organic coating, each of which could increase $N_2O_5$ uptake.

It is also important here to note that the lack of chloride enhancement in McDuffie may also contribute to the lower uptake coefficients from McDuffie compared to Yu and that Yu et al. included the chloride dependence in their parameterization specifically because they found that it better reproduced observed uptake coefficients in China.

10. Line 356 – It seems that this comparison (Base compared to NoHet) is not actually representing the full impact of $N_2O_5$/ClNO2 chemistry as indicated here. For that, the $N_2O_5$ uptake would also need to be set to zero. Instead, this comparison is showing the sensitivity of the model to the aerosol chloride enhancement of $N_2O_5$ uptake ([Cl-] = 0) and the production of ClNO2. The authors should clarify that this comparison is mainly assessing the impact of ClNO2 production, not the more general role of '$N_2O_5$-ClNO2 chemistry'.

11. Lines 459-462 – clarify that the McDuffie parameterization purposefully does not include any dependence on aerosol chloride since the exclusion of this enhancement (original proposed by Bertram and Thornton) was found to better reproduce wintertime reactive nitrogen observations in the eastern U.S. And conversely, in the previous paragraph, clarify that the Yu parameterization includes a dependence on chloride because the study authors found that this form better reproduced gamma $N_2O_5$ observations in China.

12. Line 517 – To increase study reproducibility and transparency, the authors may want to consider including a link to their model simulation code (or at least copies of the files that were changed in each sensitivity simulation).

13. Figure 4 – In addition to the maps, it would be helpful to show the correlation plots between the model and observations (perhaps as a supplemental figure).

**Technical Corrections** – suggested changes are in *blue italics*

Line 15 – Also note the impact of this chemistry on $PM_{2.5}$ in addition to $O_3$ (since this is one of the air pollutants you investigate in this study).

Line 19 – Change '…as well as their sensitivities to…' to '…as well as *the sensitivity of air pollution formation* to…'

Line 20-22 – Suggest changing this sentence to improve clarity, for example: "*Model simulations are evaluated against multiple observational datasets across China and show significant improvement in reproducing observations of particulate chloride, $N_2O_5$, and ClNO2 when including anthropogenic chlorine emissions and updates to the parameterization of $N_2O_5$-ClNO2 chemistry relative to the default model*."

Line 23 – define MDA8 here, not on line 29.

Lines 22-33. Make sure to specify that the model 'simulations' show changes in pollutants concentrations. For example, the sentence on line 22 could say, "*Model simulations show that* total tropospheric chlorine chemistry could increase annual mean MDA8 $O_3$…". Similarly, on line 28, update to say "With the additional chlorine emissions, *simulations show that* annual mean MDA8 $O_3$ in China would increase by up to…"

Line 27 – Change to "seen ozone underestimations *relative to observations*."

Line 58 – Provide a reference for this statement.

In the introduction – The authors could also cite (Simpson et al., 2015) or (Saiz-Lopez and von Glasow, 2012) as reviews of chlorine chemistry in the troposphere.

Line 67 – Change this sentence to more explicitly state that previous global and hemispheric models found that ClNO2 formation could impact ozone. Not just that it was 'suggested'.

Line 76 – Change to '*There are two key parameters that determine the uptake efficiency of $N_2O_5$ and production ClNO2*, the aerosol uptake coefficient of $N_2O_5$ (gamma) and the ClNO2 yield (phi)."

Line 112 – The doi of the 12.9.3 version should also be included here, as per GEOS-Chem recommendations (https://geos-chem.seas.harvard.edu/narrative).

Line 119 – It is also appropriate to cite (Wang et al., 2019) here since the recent updates to the model halogen chemistry are described in that paper.

Line 124 – The reference to (Wang et al., 2019) that is listed here does not appear in the reference list at the end.

Line 139 – The 75% scaling factor as implemented in GEOS-Chem is actually from (McDuffie et al., 2018a), not Lee et al., 2018. This reference should be updated.

Line 143 -145– This sentence is not quite correct as $N_2O_5$ uptake and ClNO2 yield were not directly observed in this study. It is more accurate to say here that '*The coefficients for the parameterizations in Eq. 1 and E. 2 were derived from applying a box model to observations of $N_2O_5$, ClNO2, $O_3$, and NOx mixing ratios during the winter in the eastern U.S. However, there are large uncertainties in both the values of the coefficients and functional form of the parameterizations, specifically related to their applicability to other regions*.'

Line 198 – What do the authors mean by '…could be up to…'? Do the authors mean, '…are up to…'?

Line 214 – Remove 'improved' here since the chemistry is the same in both parameterizations and at this point in the text, the Yu vs. McDuffie parameterizations have not been evaluated. Suggest changing to "… *as well as $N_2O_5$ uptake and ClNO2 production represented by the Yu parameterizations*."

Line 262 – The authors could consider moving the NMB results to this sentence to more easily compare with the NoEm case. E.g., *0.77 +/- 0.54 (NMB 39%), 0.71 +/- 0.52 (NMB -36%), and 4.5 +/- 2.4 ug m-3 (NMB -4.7%)*.

Line 287 – Replace 'The comparison indicates…' with '*The comparison between the McDuffie and Base simulations indicate*...". It is also important to clarify that this evaluation is specific to China and that differences between the Yu and McDuffie parameterizations have not been evaluated elsewhere.

Line 386 – Specify which simulations are being compared in this paragraph (and Figure 5) (e.g., the Base and NoAll simulations?)

Line 442 – change 'seas' to '*sea'*

Figure 1 – in the figure caption, define the 5 regions highlighted in panel A.

Figure 3. In the figure caption, note that the simulation definitions are provided in Table 2.

**References**
Anttila, T., Kiendler-Scharr, A., Tillmann, R., and Mentel, T. F.: On the Reactive Uptake of Gaseous Compounds by Organic-Coated Aqueous Aerosols: Theoretical Analysis and Application to the Heterogeneous Hydrolysis of N2O5, The Journal of Physical Chemistry A, 110, 10435-10443, 10.1021/jp062403c, 2006.

Bertram, T. H., and Thornton, J. A.: Toward a general parameterization of N2O5 reactivity on aqueous particles: the competing effects of particle liquid water, nitrate and chloride, Atmos. Chem. Phys., 9, 8351-8363, 10.5194/acp-9-8351-2009, 2009.

McDuffie, E. E., Fibiger, D. L., Dubé, W. P., Lopez Hilfiker, F., Lee, B. H., Jaeglé, L., Guo, H., Weber, R. J., Reeves, J. M., Weinheimer, A. J., Schroder, J. C., Campuzano-Jost, P., Jimenez, J. L., Dibb, J. E., Veres, P., Ebben, C., Sparks, T. L., Wooldridge, P. J., Cohen, R. C., Campos, T., Hall, S. R., Ullmann, K., Roberts, J. M., Thornton, J. A., and Brown, S. S.: ClNO$_2$ Yields From Aircraft Measurements During the 2015 WINTER Campaign and Critical Evaluation of the Current Parameterization, J. Geophys. Res. Atmos., 123, 12,994-913,015, 10.1029/2018JD029358, 2018a.

McDuffie, E. E., Fibiger, D. L., Dubé, W. P., Lopez-Hilfiker, F., Lee, B. H., Thornton, J. A., Shah, V., Jaeglé, L., Guo, H., Weber, R. J., Michael Reeves, J., Weinheimer, A. J., Schroder, J. C., Campuzano-Jost, P., Jimenez, J. L., Dibb, J. E., Veres, P., Ebben, C., Sparks, T. L., Wooldridge, P. J., Cohen, R. C., Hornbrook, R. S., Apel, E. C., Campos, T., Hall, S. R., Ullmann, K., and Brown, S. S.: Heterogeneous N$_2$O$_5$ Uptake During Winter: Aircraft Measurements During the 2015 WINTER Campaign and Critical Evaluation of Current Parameterizations, J. Geophys. Res. Atmos., 123, 4345-4372, 10.1002/2018JD028336, 2018b.

Morgan, W. T., Ouyang, B., Allan, J. D., Aruffo, E., Di Carlo, P., Kennedy, O. J., Lowe, D., Flynn, M. J., Rosenberg, P. D., Williams, P. I., Jones, R., McFiggans, G. B., and Coe, H.: Influence of aerosol chemical composition on $N_2O_5$ uptake: airborne regional measurements in northwestern Europe, Atmos. Chem. Phys., 15, 973-990, 10.5194/acp-15-973-2015, 2015.

Saiz-Lopez, A., and von Glasow, R.: Reactive halogen chemistry in the troposphere, Chemical Society Reviews, 41, 6448-6472, 10.1039/C2CS35208G, 2012.

Simpson, W. R., Brown, S. S., Saiz-Lopez, A., Thornton, J. A., and von Glasow, R.: Tropospheric Halogen Chemistry: Sources, Cycling, and Impacts, Chemical Reviews, 115, 4035-4062, 10.1021/cr5006638, 2015.

Wang, X., Jacob, D. J., Eastham, S. D., Sulprizio, M. P., Zhu, L., Chen, Q., Alexander, B., Sherwen, T., Evans, M. J., Lee, B. H., Haskins, J. D., Lopez-Hilfiker, F. D., Thornton, J. A., Huey, G. L., and Liao, H.: The role of chlorine in global tropospheric chemistry, Atmos. Chem. Phys., 19, 3981-4003, 10.5194/acp-19-3981-2019, 2019.

Yu, C., Wang, Z., Xia, M., Fu, X., Wang, W., Tham, Y. J., Chen, T., Zheng, P., Li, H., Shan, Y., Wang, X., Xue, L., Zhou, Y., Yue, D., Ou, Y., Gao, J., Lu, K., Brown, S. S., Zhang, Y., and Wang, T.: Heterogeneous N2O5 reactions on atmospheric aerosols at four Chinese sites: improving model representation of uptake parameters, Atmos. Chem. Phys., 20, 4367-4378, 10.5194/acp-20-4367-2020, 2020.

---

## Author Comment (AC2)

**We thank the reviewers for their supportive and thoughtful comments. Our responses to the comments are provided below, with the reviewers' comments italicized.**

Review 1:

**General Comments:**

*In this manuscript, the authors present an analysis that uses simulations with the GEOS-Chem model to assess the sensitivity of air quality (e.g., $PM_{2.5}$ and $O_3$) in China to changes in tropospheric chlorine chemistry, chlorine emissions, and the parameterization of heterogeneous $N_2O_5/ClNO_2$ chemistry. This study also provides an important evaluation of the inclusion of chlorine emissions and updated parameterizations for heterogenous $N_2O_5/ClNO_2$ chemistry, showing that both updates better reproduce surface observations of $N_2O_5$, $ClNO_2$, and particulate chloride as compared to the default GEOS-Chem model performance. This is a well-designed study and well-written manuscript, though I have specific comments and suggestions below to help deepen the discussion and improve the clarity of the text. For example, there are a few cases where it would help if the authors could be more specific about the processes they are referring to when they say 'N₂O₅-ClNO₂ chemistry'. A more detailed discussion on the differences and similarities in the heterogenous parameterizations would also help the reader better interpret the simulation results (see more comments below).*

*Overall, this study is an important contribution to the field as it highlights that simulations of absolute air pollutant concentrations are sensitive to both chloride emissions and the parameterization of heterogeneous $N_2O_5$ and $ClNO_2$ chemistry. It also shows that the model's sensitivity to emissions depends on these heterogenous parameterizations. Not only do these results help improve our understanding of the importance of these processes, but they also show that air quality impacts from anthropogenic chlorine sources should be considered in air quality improvement strategies, and that our ability to accurately predict the magnitude of the reduction benefits will depend on the modeled representation of heterogeneous $N_2O_5$ chemistry. I recommend that this study be accepted for publication after the specific comments below are addressed.*

Thanks for the supportive and helpful comments. We have addressed all the concerns raised by the reviewer, including the term of "N₂O₅-ClNO₂ chemistry" and "the differences and similarities in the heterogenous parameterizations" (e.g. the replies to the general comments #2 and #1,

respectively). Please see below for the point-by-point response to the reviewer's comments and concerns.

General Comments:

1. *This comment applies to all sections of the manuscript. In cases where the authors simply state that one parameterization performs better than another, it would be helpful if the authors could provide additional discussions on why one might be outperforming the other in terms of the processes controlling $N_2O_5$ uptake and $ClNO_2$ yield. For example, the Yu and McDuffie parameterizations for $N_2O_5$ uptake are actually based on the same general parameterization (see specific comment 4), but the differences are that the Yu parameterization does not consider added suppression from organic aerosol (included in McDuffie), includes an uptake enhancement from particulate chloride (not included in McDuffie), and uses different rate coefficient ratios than McDuffie. Therefore, since this study shows that the Yu parameterization is able to better reproduce available surface observations, the results here suggest that in China, it may be important to consider particulate chloride in the uptake of $N_2O_5$ and that organics may not play as important of a suppressive role as shown in previous studies. Including these details helps explain to the reader why one parameterization may be outperforming another and provides insight into specific modeled processes that could be improved. Many of my specific comments below are related to this general comment.*

  Thanks for the constructive comment. We agree with the reviewer that compared with the Yu parameterization, the underestimation of $\gamma_{N2O5}$ and $\varphi_{ClNO2}$ from the McDuffie parameterization in China is potentially due to (1) overpredicting the suppressive role of organic aerosol, (2) the lack of the uptake enhancement from particulate chloride, and (3) the scaling factor (i.e. 0.25 in Eq. 4) applied in the McDuffie parameterization. As suggested by the reviewer, we have added more discussion throughout the manuscript to make it much clearer about the difference and similarities in the Yu and McDuffie parameterizations.

  Firstly, we added more detailed description about the Yu and McDuffie parameterizations, including adding Eq. 2 and 3 for the calculation of $\gamma_{core}$ and $\gamma_{coat}$ as well as the explanation of the forms of the two parameterizations. For example, in line 145 – 176 for the McDuffie parameterization: "McDuffie parameterization is the first field-based empirical parameterizations modified

from the framework proposed by previous laboratory studies including BT09 (Anttila et al., 2006; Bertram and Thornton, 2009; Riemer et al., 2009). It is adjusted to reproduce the mean values of $\gamma_{N2O5}$ and $\varphi_{ClNO2}$ observed from ambient wintertime aircraft measurements over the eastern U.S. The parameterization for $\gamma_{N2O5}$ accounts for both the inorganic and organic aerosol components …".

In line 177 – 186 for the Yu parameterization: "Recently, Yu et al. (2020) proposed new parameterizations of $\gamma_{N2O5}$ and $\varphi_{ClNO2}$ based on BT09 to account for the dependence on aerosol water, nitrate, and chloride concentrations but with coefficients derived from uptake coefficients directly measured on ambient aerosol in two rural sites in China. The parameterizations …".

The forms of the Eq. 2 and 5 and the forms of the Eq. 4 and 6 are adjusted to be consistent with each other so that the comparison between each other is more intuitive (see more details in the reply to the specific comment #4 below).

Secondly, we added detailed discussion providing a general picture of how the Yu and McDuffie parameterizations differ from each other in line 187 – 205: "Although both the two parameterizations are developed based on BT09, there exit significant differences of $\gamma_{N2O5}$ and $\varphi_{ClNO2}$ between McDuffie and Yu parameterizations. For $\gamma_{N2O5}$, McDuffie parameterization generally follows BT09 for the calculation of the uptake on inorganic aerosols (i.e. $\gamma_{core}$), but excludes the dependence on aerosol chloride so as to better reproduce observed wintertime reactive nitrogen in eastern U.S. Moreover, the parameterization accounts for the suppressive effects of the organics (i.e. $\gamma_{coat}$), which is not directly included in BT09 (Anttila et al., 2006; Riemer et al., 2009; Morgan et al., 2015). In contrast to McDuffie parameterization, Yu parameterization excludes the organic suppression but includes the chloride enhancement so as to better reproduce $\gamma_{N2O5}$ observed in China (Yu et al., 2020). It is worth mentioning that the coefficients applied in the parameterization of $\gamma_{N2O5}$ also differ between McDuffie and Yu parameterizations as both are fixed to reproduce the ambient observation representing different pollution conditions. For example, $k_a$ is equal to 0.04 in Eq. 2 but 0.033 in Eq. 5. The $\gamma_{N2O5}$ in McDuffie parameterization is thus expected to be lower compared with the Yu parameterization due to the resistance from organic coating and the lack of the chloride enhancement. For $\varphi_{ClNO2}$, both the McDuffie and Yu parameterizations are based on BT09, but with different coefficients (i.e. $k_c = 1/450$ in Eq. 4 and 1/150 in Eq. 6). Although $k_c$ in Eq. 4 is relatively smaller, the scaling factor of

0.25 applied in Eq. 4 ultimately results in a much smaller $\varphi_{ClNO2}$ in McDuffie parameterization compared with Yu parameterization under the same condition. Again, keep it in mind that McDuffie parameterization is derived from fits to observations over the eastern U.S. (McDuffie et al., 2018a) while Yu parameterization is fitted to observations at rural locations in China (Yu et al., 2020)."

Moreover, to help explain to the reader why Yu parameterization performs better, we also modified the corresponding discussion when comparing the results of $N_2O_5$ and $ClNO_2$ from different simulation cases:

[revised manuscript text omitted]

2. *Throughout the text, the authors refer to the simulations with the Yu parameterization as simulations with 'updated $N_2O_5$-$ClNO_2$ chemistry'. This is slightly confusing terminology for the reader as it is not actually the chemical*

*reactions of N₂O₅ and ClNO₂ that are changing between the simulations. Rather, it is the parameterizations of gamma N₂O₅ and phi ClNO₂ that are changing. I would suggest editing the terminology throughout the paper to reflect this difference. E.g., Replace 'N₂O₅-ClNO₂ chemistry' with something along the lines of 'updated parameterizations for heterogeneous N₂O₅ and ClNO₂ chemistry'.*

Thanks for the comment. We have replaced the terminology "N₂O₅-ClNO₂ chemistry" by "heterogeneous $N_2O_5$ + Cl chemistry'. the term "updated N₂O₅-ClNO₂ chemistry" was also replaced by "updated parameterizations for heterogeneous $N_2O_5$ + Cl chemistry" throughout the text.

3. *The authors may also want to consider including maps or averages of model-calculated N₂O₅ uptake coefficients and ClNO₂ yields. Reporting these values could help future studies compare the results from each parameterization with available field-derived values. The model diagnostics should be able to provide these values. This could be included in the supplement.*

Thanks for the comment. We have added the maps of $\gamma_{N2O5}$ and $\varphi_{ClNO2}$ for different simulation cases in the Supplementary Material (Fig. S3 and S4). The corresponding description was also added throughout the manuscript. A few examples are listed below:

line 335 – 338 for the comparison of $\gamma_{N2O5}$ between the Base and NoEm cases: "As shown in Figure S3, although the values of $\gamma_{N2O5}$ between the Base and NoEm cases are similar over the ocean, the Base case has relatively higher $\gamma_{N2O5}$ over China compared with the NoEm case (0.016 vs. 0.014 on annual mean basis).".

Line 347 – 350 for the comparison of $\gamma_{N2O5}$ between the Base and McDuffie parameterization: "The overestimate of $N_2O_5$ in McDuffie parameterization suggests the potential underestimate in the corresponding $\gamma_{N2O5}$. As shown Figure S3, the value of $\gamma_{N2O5}$ from the McDuffie case is much smaller than that from the Base case (0.0071 vs. 0.016 averaged over China).".

line 376 – 379 for the comparison of $\varphi_{ClNO2}$: "The difference in $ClNO_2$ concentrations is mainly associated with distinct $\varphi_{ClNO2}$ values among different cases. As shown in Figure S4, the value of $\varphi_{ClNO2}$ is significantly higher in the Base case (0.36 averaged over China) than in the NoEm (0.14) and McDuffie (0.11) cases.".

**Specific Comments:**

1. *Line 80-83 – These two sentences are technically correct, but the authors should adjust them to specifically note the possible role of organics (not considered in the (Bertram and Thornton, 2009) study). It is the presence of organic aerosol species that is thought to reduce the uptake of N₂O₅ relative to that predicted by the Bertram and Thornton parameterization. It is not just the mixing state that is a possible difference, but the presence of hydrophobic organic aerosol that lead to a potentially complex aerosol mixing state.*

> Thanks for the comment. We revised the sentences into: "The most widely used parameterization for $\gamma_{N2O5}$ and $\varphi_{ClNO2}$ was proposed by Bertram and Thornton (2009) (hereinafter referred to as BT09), which is based on the laboratory studies with considerations of aerosol water content, concentrations of nitrate and chloride, and specific surface area (i.e. the ratio of surface area concentrations to particle volume concentrations). However, recent field and model studies have shown that this parameterization would overestimate both $\gamma_{N2O5}$ and $\varphi_{ClNO2}$, especially in regions with high Cl levels (Mcduffie et al., 2018b; Mcduffie et al., 2018a; Xia et al., 2019; Chang et al., 2016; Hong et al., 2020; Yu et al., 2020). The discrepancies could be partly attributed to the complexity of atmospheric aerosols (e.g. mixing state and complex coating materials) in contrast to the simple proxies used in laboratory studies (Yu et al., 2020). Specifically, the suppressive effect of organic coatings is not considered in BT09.".

2. *Line 87-89 – To strengthen this last point, the authors could mention that some of these previous field-based parameterizations were derived from observations in locations with conditions may not be applicable to the highly polluted regions in China. This makes it important to evaluate these parameterizations under different conditions. The authors should also clarify that in this study, a full evaluation of this chemistry is being conducted for China specifically. As written, 'full evaluation' makes it sound as if this study will be conducting a global analysis.*

> Thanks for the comment. We have modified the corresponding description into: "However, some of these previous field-based parameterizations were derived from observations under different ambient conditions which may not be applicable to the highly polluted regions in China. A full evaluation of the

representativeness of different parameterizations for the heterogeneous $N_2O_5$ + Cl chemistry and the associated impacts on ambient air quality in China is not available yet.".

3. *Line 93 – My comment is on the statement 'The importance of anthropogenic chlorine emissions, which were ignored in most studies…". While these emissions are not commonly included in modeling studies, there is a clear example in (Wang et al., 2019) where they did not 'ignore' chlorine emissions but rather found that the addition of anthropogenic chlorine emissions in GEOS-Chem resulted in overestimates of HCl observations in the U.S. This is an important study to cite in this section. The authors could still note that anthropogenic emissions in China may be relatively more important in China than in the U.S., which is why it is important to study their impacts here.*

We agree with the reviewer that the study by Wang et al. (2019) has tested anthropogenic chlorine emissions in the simulation but found insignificant influence of anthropogenic Cl emission in the U.S. To avoid misleading, we have revised the corresponding description into: "In early modelling studies, global tropospheric chlorine is mainly from sea salt aerosols (SSA), and most of the chlorine over continental regions in North America and Europe is dominated by the long-range transport of SSA (Wang et al., 2019; Sherwen et al., 2017). The study by Wang et al. (2019) found an addition of anthropogenic chlorine emissions in the model would result in overestimates of HCl observations in the U.S and suggested insignificant influence of anthropogenic Cl in the U.S. However, there are also studies pointing out the importance of anthropogenic chlorine emissions in China (Le Breton et al., 2018; Yang et al., 2018; Hong et al., 2020). The study by Wang et al. (2020b) suggested that anthropogenic chlorine emissions in China are more than 8 times higher than those in the U.S., and could dominate reactive chlorine in China, resulting in an increase in $PM_{2.5}$ and Ozone by up to 3.2 $\mu g\ m^{-3}$ and 1.9 ppbv on annual mean basis, respectively.".

4. *Lines 131 - 154 – Per my general comment above, it would be helpful in this section for the authors to provide a more detailed narrative on 1) the differences between the various parameterizations, 2) the processes that they are trying to represent, and 3) a brief summary of how each parameterization was derived.*

*For example, for (1 and 2), the authors should include in Eq. 1 the detailed equations for both gamma_coat and gamma_core (found in (McDuffie et al., 2018b)). These details are important to include here because the functional form of the Yu parameterization is actually the same as that of gamma_core from the McDuffie parameterization. By showing the full equations, the readers can better understand that the only differences between the Yu and McDuffie uptake parameterizations are that Yu parameterization includes an enhancement from aerosol chloride (not included in McDuffie), does not consider the added resistance from an organic coating (included in McDuffie) and uses different coefficients (reaction rate constant ratios) than McDuffie in the gamma_core equation.*

*For (3), to better understand why the parameterizations are different, it would be useful to briefly explain that the form of the McDuffie parameterization is derived from parameterizations proposed in multiple laboratory studies ((Bertram and Thornton, 2009), Riemer et al., 2009, and (Anttila et al., 2006)) to account for the uptake dependence on aerosol water and nitrate concentrations and added resistance from an organic aerosol coating. The coefficients in this parameterization were then derived by fitting a chemical box model to aircraft observations of $N_2O_5$, $ClNO_2$, $O_3$, and $NO_x$ during the winter over the eastern U.S. In contrast, for the Yu parameterization, it would be important to note that the form of this parameterization is from Bertram and Thornton to account for the dependence on aerosol water, nitrate, and chloride concentrations, with coefficients derived from uptake coefficients directly measured on ambient aerosol in two rural sites in China. It would also be important to note that while previous studies have found that organics can suppress $N_2O_5$ uptake, (Yu et al., 2020) found that including the chloride enhancement and excluding the organic coating best reproduced the observed $N_2O_5$ uptake coefficients in their study.*

*Similarly for Eq 2., replace k2/k3 with 1/450 to make this form consistent with Eq. 4. Also add a note that both forms are from Bertram and Thornton, again with coefficients and scaling factors (in the case of McDuffie) derived from fits to observations over the eastern U.S. (McDuffie) and rural locations in China (Yu).*

Thanks for the constructive comment. As replied to the general comment #1, we have modified the whole section (Section 2.1.1) to provide a more detailed description about the similarity and difference between these two parameterizations. Specifically, for (1 and 2), we have added Eq. 2 and 3 for the

calculation of $\gamma_{core}$ and $\gamma_{coat}$, respectively. The forms of the Eq. 2 and 5 as well as the forms of the Eq. 4 and 6 are adjusted to be consistent with each other so that the comparison between each other is more intuitive. For example, we replaced $k_2/k_3$ with $k_c$ in Eq. 4 and 6. We also used $k_a$ in Eq. 2 and 5 as the rate constant ratio representing the competition between aerosol-phase $H_2O$ and $NO_3^-$ for the $H_2ONO_2^+(aq)$ intermediate. For (3), as suggested by the reviewer, we have emphasized that "McDuffie parameterization is the first field-based empirical parameterization derived from the framework proposed in multiple laboratory studies including BT09 (Anttila et al., 2006; Bertram and Thornton, 2009; Riemer et al., 2009) to account for the uptake dependence on aerosol water and nitrate concentrations as well as the resistance from an organic coating. The coefficients for McDuffie parameterization were derived from applying a box model to observations of $N_2O_5$, $ClNO_2$, $O_3$, and $NO_x$ mixing ratios during the winter in the eastern U.S." and that the Yu parameterization is "based on BT09 to account for the dependence on aerosol water, nitrate, and chloride concentrations but with coefficients derived from uptake coefficients directly measured on ambient aerosol in two rural sites in China." Please see more detailed discussion in line 142-205 in Section 2.1.1.

5. *Line 183 – It would be helpful context for the reader to specify which sectors contribute to emissions of HCl, $Cl_2$, and particulate chloride in the inventories.*

     Thanks for the comment. We have added Table S1 in the Supplementary Material to specify the contribution from different sectors to anthropogenic emissions of HCl, $Cl_2$, and $Cl^-$.

6. *Line 221 – The authors should clarify that the NoHet case not only sets phi $ClNO_2$ to zero, but also removes the enhancement of $N_2O_5$ uptake from aerosol chloride. So this simulation actually tests the model sensitivities to a smaller gamma $N_2O_5$ and zero $ClNO_2$ production.*

     Thanks for the comment. To avoid confusion, we have revised the corresponding description into: "In addition, while keeping others the same as the Base case, the NoHet case sets $\varphi_{ClNO2}$ to zero (Eq.6) and removes the enhancement of $N_2O_5$ uptake from aerosol chloride (i.e. $[Cl^-] = 0$ in Eq. 5). The comparison between the Base and NoHet cases could thus evaluate the

importance of the heterogeneous $N_2O_5$ $+$ Cl chemistry (i.e., the model sensitivities to a smaller gamma $N_2O_5$ and zero $ClNO_2$ production).".

7. *Line 278 – The comparison here is between the Base and NoEm cases. Therefore, it seems that the results imply that additional chlorine emissions could increase the uptake coefficient due to increased aerosol chloride. This comparison does not directly evaluate the Yu parameterization as implied, since the Yu parameterization is included in both Base and NoEm simulations. This sentence should be updated accordingly.*

To make it clear, we have revised the sentence into: "The improvement in the Base case is apparent at most sites, implying that additional chlorine emissions could effectively increase the uptake coefficient of $N_2O_5$ in Yu parameterization.".

8. *Line 290 – There are many other studies (see section 4.2.6 in (McDuffie et al., 2018b)) that have shown that organics can suppress uptake, which should also be referenced here (the authors can still note that Yu found that excluding the organic coating best reproduced uptake coefficients observed in China). In addition, (Morgan et al., 2015) actually state that "An additional suppression of the parameterised ($N_2O_5$) uptake is likely required to fully capture the variation in $N_2O_5$ uptake, which could be achieved via the known suppression by organic aerosol. However, existing parameterisations representing the suppression by organic aerosol were unable to fully represent the variation in $N_2O_5$ uptake." Therefore, the sentence should be amended to clarify that organic suppression may be important to consider in the estimate of $N_2O_5$ uptake, but that the currently implemented parameterization may overpredict the level of suppression.*

To avoid misleading, we have revised the corresponding discussion into: "The underestimate in $\gamma_{N2O5}$ from the McDuffie case could to large extent be explained by the suppressive effect of organic coatings ($\gamma_{coat}$) as discussed above in Section 2.1.1. The magnitude of the organic suppression is highly dependent on many factors (e.g. organic composition, particle phase state, etc.) and thus remains poorly quantified (Griffiths et al., 2009; Gross et al., 2009; Thornton et al., 2003). Although many studies have shown that organic aerosol can suppress the $N_2O_5$ uptake (Anttila et al., 2006; Riemer et al., 2009), the level of organic

suppression may be overpredicted in currently implemented parameterization attributed to the poorly quantified and/or unknown factors (e.g. Morgan et al. (2015)). For example, some studies found that ignoring the difference between water-soluble and water-insoluble organics may lead to an upper limit for the suppressive effect of organic coatings and consequently an underestimate in the solubility and diffusivity of $N_2O_5$ in organic matter (Chang et al., 2016; Yu et al., 2020). Although the $\gamma_{coat}$ in McDuffie parameterization is calculated as a function of organic aerosol O:C ratio and RH (see Eq. 2), which could increase with higher RH and higher O:C ratio, it may still overpredict the suppressive role of organic coatings in China. On the other hand, the study by Yu et al. (2020) found that excluding the organic coating best reproduced uptake coefficients observed in China. In addition, the underestimate in $\gamma_{N2O5}$ in McDuffie parameterization in China could also be to some extent explained by the lack of the chloride enhancement (also discussed in Section 2.1.1). It is worth noting that the evaluation here is specific to China and the differences between Yu and McDuffie parameterizations have not been evaluated elsewhere.".

9. *Line 293 – 297 – The McDuffie parameterization is slightly more sophisticated than indicated in this sentence. For example, in the McDuffie parameterization, the gamma_coat value is actually calculated as a function of organic aerosol O:C ratio and RH. These factors are meant to account for conditions where higher relative humidity and higher O:C ratio may represent less likely liquid-liquid aerosol phase separation, a partially coated aerosol, or thinner organic coating, each of which could increase $N_2O_5$ uptake.*

*It is also important here to note that the lack of chloride enhancement in McDuffie may also contribute to the lower uptake coefficients from McDuffie compared to Yu and that Yu et al. included the chloride dependence in their parameterization specifically because they found that it better reproduced observed uptake coefficients in China.*

Thanks for the comment. As replied to the specific comment #8, we have modified the discussion about the underestimate of $N_2O_5$ uptake in China from McDuffie parametrization. Specifically, we have emphasized that the suppression effect in McDuffie parametrization is a function of RH and O:C ratio and added the discussion about the lack of chloride enhancement in

McDuffie parametrization in line 361 – 366: "Although the $\gamma_{coat}$ in McDuffie parametrization is calculated as a function of organic aerosol O:C ratio and RH (see Eq. 2), which could increase with higher RH and higher O:C ratio, it may still overpredict the suppressive role of organic coatings in China. On the other hand, the study by Yu et al. (2020) found that excluding the organic coating best reproduced uptake coefficients observed in China. In addition, the underestimate in $\gamma_{N2O5}$ in McDuffie parameterization in China could also be to some extent explained by the lack of the chloride enhancement (also discussed in Section 2.1.1).".

10. *Line 356 – It seems that this comparison (Base compared to NoHet) is not actually representing the full impact of $N_2O_5/ClNO_2$ chemistry as indicated here. For that, the $N_2O_5$ uptake would also need to be set to zero. Instead, this comparison is showing the sensitivity of the model to the aerosol chloride enhancement of $N_2O_5$ uptake ([Cl⁻] = 0) and the production of $ClNO_2$. The authors should clarify that this comparison is mainly assessing the impact of $ClNO_2$ production, not the more general role of '$N_2O_5$-$ClNO_2$ chemistry'.*

As replied to the specific comment #6, we have revised the description of the NoHet case into: "The comparison between the Base and NoHet cases could thus evaluate the importance of the of heterogeneous $N_2O_5$ ＋ Cl chemistry (i.e., the model sensitivities to a smaller gamma $N_2O_5$ and zero $ClNO_2$ production". To make it clear, we also modified the description in line 441 – 445 to clarify the comparison is mainly for the impact of $ClNO_2$ production instead of the more general role of the heterogenous chemistry of $N_2O_5$: "Therefore, we further investigate the role that the heterogeneous $N_2O_5$ ＋ Cl chemistry plays in tropospheric chlorine chemistry through the comparison between the Base and NoHet (Fig. 6 and Fig. S8) cases. Keep it in mind that the comparison is mainly assessing the impact of $ClNO_2$ production, namely the uptake of $N_2O_5$ on chloride aerosol, not the general role of $N_2O_5$ heterogeneous chemistry.".

11. *Lines 459-462 – clarify that the McDuffie parameterization purposefully does not include any dependence on aerosol chloride since the exclusion of this enhancement (original proposed by Bertram and Thornton) was found to better*

*reproduce wintertime reactive nitrogen observations in the eastern U.S. And*
*conversely, in the previous paragraph, clarify that the Yu parameterization*
*includes a dependence on chloride because the study authors found that this form*
*better reproduced gamma $N_2O_5$ observations in China.*

Thanks for the comments. As replied to the general comment #1, we have added detailed discussion about the similarity and difference between these two parameterizations, including whether the effects of organic aerosol and chloride aerosol are included or excluded in the parameterizations. Also, as suggested by the reviewer, we added one sentence in line 558 – 560: "This is consistent with the dependence on chloride in Yu parameterization, which is included to better reproduce $\gamma_{N2O5}$ observations in China (Yu et al., 2020)." and also modified the text in line 565 – 567 into: "This insensitivity to chlorine emissions could be expected from Eq. 2 where the dependence on aerosol chloride is not included so as to better reproduce wintertime reactive nitrogen observations in the eastern U.S.".

12. *Line 517 – To increase study reproducibility and transparency, the authors may want to consider including a link to their model simulation code (or at least copies of the files that were changed in each sensitivity simulation).*

Thanks for the comments. We have uploaded the revised code for each sensitivity simulation and also added the following sentence here: "The revised codes for different simulations could be downloaded via https://zenodo.org/record/5957287#.YfyNMppBxPZ".

13. *Figure 4 – In addition to the maps, it would be helpful to show the correlation plots between the model and observations (perhaps as a supplemental figure).*

Thanks for the comments. We have added correlation maps of MAD8 $O_3$ and $PM_{2.5}$ between the model and observations as Figure S6 in the Supplementary Material.

**Technical Corrections – suggested changes are in *blue italics***

1. *Line 15 – Also note the impact of this chemistry on PM 2.5 in addition to O 3 (since this is one of the air pollutants you investigate in this study).*

    Fixed!

2. *Line 19 – Change '...as well as their sensitivities to…' to '...as well as the sensitivity of air pollution formation to…'*

    Done!

3. *Line 20-22 – Suggest changing this sentence to improve clarity, for example: "Model simulations are evaluated against multiple observational datasets across China and show significant improvement in reproducing observations of particulate chloride, N2O5, and ClNO2 when including anthropogenic chlorine emissions and updates to the parameterization of N2O5-ClNO2 chemistry relative to the default model."*

    Done!

4. *Line 23 – define MDA8 here, not on line 29.*

    Fixed!

5. *Lines 22-33. Make sure to specify that the model 'simulations' show changes in pollutants concentrations. For example, the sentence on line 22 could say, "Model simulations show that total tropospheric chlorine chemistry could increase annual mean MDA8 O3 …". Similarly, on line 28, update to say "With the additional chlorine emissions, simulations show that annual mean MDA8 O3 in China would increase by up to…"*

    Fixed!

6. *Line 27 – Change to "seen ozone underestimations relative to observations."*

    Done!

7. *Line 58 – Provide a reference for this statement.*

   Done!

8. *In the introduction – The authors could also cite (Simpson et al., 2015) or (Saiz-Lopez and von Glasow, 2012) as reviews of chlorine chemistry in the troposphere.*

   Thanks for the comments! As suggested by the reviewer, the references are now cited in line 46 – 48: "In general, Cl atom can be produced from the photo-dissociation and the oxidation of chlorinated organic species (e.g. $CH_3Cl$, $CH_2Cl_2$ and $CHCl_3$) and inorganic chlorine species (i.e. HCl and $Cl_2$) (Saiz-Lopez and Von Glasow, 2012; Simpson et al., 2015)." and also in line 48 – 52: "Nitryl chloride ($ClNO_2$), formed through the heterogeneous reaction between dinitrogen pentoxide ($N_2O_5$) and chloride-containing aerosols (hereinafter referred to as the heterogeneous $N_2O_5$ ＋ Cl chemistry), is found to be another important source of tropospheric Cl atoms in polluted regions (Liu et al., 2018; Haskins et al., 2019; Choi et al., 2020; Simpson et al., 2015).".

9. *Line 67 – Change this sentence to more explicitly state that previous global and hemispheric models found that ClNO2 formation could impact ozone. Not just that it was 'suggested'.*

   As suggested by the reviewer, we modified the sentence into: "Previous global and hemispheric models found that the heterogeneous $N_2O_5$ ＋ Cl chemistry could increase monthly mean values ..."

10. *Line 76 – Change to 'There are two key parameters that determine the uptake efficiency of $N_2O_5$ and production ClNO2, the aerosol uptake coefficient of $N_2O_5$ (gamma) and the ClNO2 yield (phi)."*

    Done!

11. *Line 112 – The doi of the 12.9.3 version should also be included here, as per GEOS-Chem recommendations (https://geos-chem.seas.harvard.edu/narrative).*

Fixed.

12. *Line 119 – It is also appropriate to cite (Wang et al., 2019) here since the recent updates to the model halogen chemistry are described in that paper.*

Done!

13. *Line 124 – The reference to (Wang et al., 2019) that is listed here does not appear in the reference list at the end.*

Fixed!

14. *Line 139 – The 75% scaling factor as implemented in GEOS-Chem is actually from (McDuffie et al., 2018a), not Lee et al., 2018. This reference should be updated.*

Updated!

15. *Line 143 -145– This sentence is not quite correct as $N_2O_5$ uptake and $ClNO_2$ yield were not directly observed in this study. It is more accurate to say here that 'The coefficients for the parameterizations in Eq. 1 and E. 2 were derived from applying a box model to observations of $N_2O_5$, $ClNO_2$, $O_3$, and $NO_x$ mixing ratios during the winter in the eastern U.S. However, there are large uncertainties in both the values of the coefficients and functional form of the parameterizations, specifically related to their applicability to other regions.'*

Thanks for the suggestion. We have revised the corresponding discussion into: "For more detailed description of McDuffie parameterization, readers are referred to McDuffie et al. (2018b; 2018a). Keep it in mind that the coefficients for the parameterizations in Eq. 1 – 4 were derived to better reproduce wintertime observations in the eastern U.S. However, there are large uncertainties in both the values of the coefficients and functional form of the parameterizations, specifically related to their applicability to other regions."

16. *Line 198 – What do the authors mean by '...could be up to…'? Do the authors mean, '...are up to…'?*

> Yes. We now replaced "could be up to" with "are up to".

17. *Line 214 – Remove 'improved' here since the chemistry is the same in both parameterizations and at this point in the text, the Yu vs. McDuffie parameterizations have not been evaluated. Suggest changing to "… as well as $N_2O_5$ uptake and $ClNO_2$ production represented by the Yu parameterizations."*

> Fixed!

18. *Line 262 – The authors could consider moving the NMB results to this sentence to more easily compare with the NoEm case. E.g., 0.77 +/- 0.54 (NMB 39%), 0.71 +/- 0.52 (NMB -36%), and 4.5 +/- 2.4 ug m-3 (NMB -4.7%).*

> Done!

19. *Line 287 – Replace 'The comparison indicates…' with 'The comparison between the McDuffie and Base simulations indicate...". It is also important to clarify that this evaluation is specific to China and that differences between the Yu and McDuffie parameterizations have not been evaluated elsewhere.*

> Done! As suggested by the reviewer, we also added the following sentence in line 366 – 368: "It is worth noting that the evaluation here is specific to China and the differences between Yu and McDuffie parameterizations have not been evaluated elsewhere."

20. *Line 386 – Specify which simulations are being compared in this paragraph (and Figure 5) (e.g., the Base and NoAll simulations?)*

> We modified the corresponding text into: "the effect of tropospheric chlorine chemistry without the heterogeneous $N_2O_5$ ＋ Cl chemistry is much smaller (Fig. S10, the comparison between the NoHet and NoChem cases) ..." Similar modification has also been made in the captions of the figures.

*21. Line 442 – change 'seas' to 'sea'*

Done!

*22. Figure 1 – in the figure caption, define the 5 regions highlighted in panel A.*

Done!

*23. Figure 3. In the figure caption, note that the simulation definitions are provided in Table 2.*

Done!

---

## Author Comment (AC3)

**We thank the reviewers for their supportive and thoughtful comments. Our responses to the comments are provided below, with the reviewers' comments italicized.**

**Review 2:**

*This paper analyses the ability of the GEOS-Chem model, run in a regional configuration, to simulate the concentration of Cl-, N2O5, ClNO2, O3 and PM under different assumptions about the anthropogenic emissions of HCl and Cl2, and for the parameterization of the heterogenous N2O5->HNO3, ClNO2 processes over China. In general the paper provides a good assessment of model performance against a good set of observations and explores the limitations of the current generation of parameterizations.*

*I have some comments about some aspects of the paper (described below) but in general I am supportive of publications if these aspects can be addressed.*

Thanks for the supportive and helpful comments. We have addressed all the concerns raised by the reviewer, including the impacts on $NO_y$ and OH. Please see below for the point-by-point response to the reviewer's comments and concerns.

*My major comment is on how the model is being analysed. Much is made of the impacts on NOx of the production of ClNO2. I think more could be made on the impacts on NOy. The reactions involved conserve NOy on some levels (if NO3- is included in the definition) but they should change the partitioning of NOy from HNO3 into more reactive forms (ClNO2, NOx etc). This is one key way in that this chemistry influences the composition - the other in the production of Cl atoms. It would be informative to look at how the fraction of NOy as NOx (if we include ClNO2 in our NOx definition) increases with the inclusion of the new chemistry. It would be useful to explore the ratio of NOx to NOy, NO3- to NOy etc with and without the chlorine chemistry switched on.*

Thanks for the constructive comment. We agree with the reviewer that the formation of $ClNO_2$ also affect the portioning of $NO_y$ from $HNO_3$ into more reactive forms. Therefore, we have added the analysis on the impacts of the heterogeneous $N_2O_5$ ＋ Cl chemistry on the partitioning of $NO_y$. The maps of the changes in the ratios of $NO_x$ to $NO_y$ and $NO_3^-$ to $NO_y$ are now provided as Fig. S9 in the supplement. The corresponding discussions are added in line 452

– 462: "In addition to the production of Cl atoms, the $ClNO_2$ formation also affects the partitioning of $NO_y$ from $HNO_3$ into more reactive forms (e.g., $NO_x$ and $ClNO_2$) through the recycling of $NO_x$, and therefore of great importance in atmospheric chemistry (Bertram et al., 2013; Li et al., 2016; Wang et al., 2020a). To analyze the impact of the heterogeneous $N_2O_5$ + Cl chemistry on $NO_y$ partitioning, Figure S9 shows the change in the ratios of $NO_x$ to $NO_y$ and $NO_3^-$ to $NO_y$ as the difference between the Base and NoHet cases. Since $ClNO_2$ could be treated as a reservoir for reactive nitrogen at night, we include $ClNO_2$ as part of $NO_x$ in the calculation ($NO_x = NO + NO_2 + ClNO_2$ and $NO_y = NO + NO_2 + ClNO_2 + HNO_3 + 2 \times N_2O_5 + NO_3 + HONO + HNO_4 + NO_3^- +$ various organic nitrates). The results show that due to the $ClNO_2$ production, the ratios of $NO_x$ to $NO_y$ increase by 1.8% averaged in China and up to 5.4% in the Sichuan Basin, Northeast Plain and North China Plain on annual mean basis. Meanwhile, the ratios of $NO_3^-$ to $NO_y$ decrease by 1.1% averaged in China and up to 5.1% in the Sichuan Basin on annual mean basis.".

*Similarily the authors sugges that the increased Cl leads to increased VOC oxidation. Perhaps some figures to explore this might also be useful and provide some evidence for their inferences on the impact on OH?*

Thanks for the suggestion. We have added the map of the changes in $HO_2$ from different simulation cases as Fig. S7 (a) in the supplement accordingly. The map of the changes in OH is already provided in Fig. S7 (b). We have also added more discussion about the impact on OH and $HO_2$ in line 425 – 435: "The increased Cl atoms could react with VOCs (especially alkanes) producing more peroxy radicals, including organic peroxy radicals ($RO_2$) and hydroperoxyl radicals ($HO_2$). As shown in Figure S7 (a), the chlorine chemistry could increase annual mean $HO_2$ concentrations by $1.6 \times 10^6$ molec cm$^{-3}$ averaged in China (up to $8.6 \times 10^6$ molec cm$^{-3}$ in the coastal regions). In the presence of NO, the peroxy radicals recycle OH while oxidize NO to $NO_2$. The subsequent photolysis of $NO_2$ could further lead to more $O_3$ production and consequently also more OH (Osthoff et al., 2008; Riedel et al., 2014; Simpson et al., 2015). On the other hand, the recycling of $NO_x$ back into the atmosphere associated with the photolysis of $ClNO_2$ could also lead to more $O_3$ production. The results here show a significant increase in surface annual mean OH (Fig. S7 (b)) and MDA8 $O_3$ (Fig. 5c) by $3.8 \times 10^4$ molec cm$^{-3}$ and 1.1 ppbv

respectively averaged in China (up to $1.2 \times 10^5$ molec cm$^{-3}$ and 4.5 ppbv respectively in the Sichuan Basin)."

**Minor Comments:**

1.  *I don't find the phase "N2O5-ClNO2 Chemistry" sitting well with my ear. It is basically one reaction (or two if you include the photolysis) in the scheme (reaction R3) and so describing it as "N2O5-ClNO2 chemistry" makes it sound like something more different. I would suggest something like the "parameterization of gN2O5" might better reflect what is happening.*

    Thanks for the comment. We have replaced "$N_2O_5$-$ClNO_2$ chemistry" with "heterogeneous $N_2O_5$ + Cl chemistry" to specifically refer to the reaction between $N_2O_5$ and chloride-containing aerosols. Besides, "updated $N_2O_5$-$ClNO_2$ chemistry" is also replaced with "updated parameterizations for the heterogeneous $N_2O_5$ + Cl chemistry" throughout the manuscript.

2.  *Line 45. I'm not sure that "Recently" describes the literature of ClNO2. The original experimental paper Finlayson-Pitts et al., 1989 probably doesn't classify are recent and there have been a number of papers from the late 2000s which describe much of this matierla.*

    Sorry for the misleading. We now delete the word "Recently".

3.  *Line 56. Can the products of the reaction been moved to the right to make it clear that this is a single reaction? It would probably be normal to include some of the products on the same line as the reactants to make this clear.*

    Fixed.

4.  *Line 76. I think the word including should be replaced by a comma.*

    Thanks for the comment. We revised the sentence into: "There are two key parameters that determine the uptake efficiency of $N_2O_5$ and production of $ClNO_2$, the aerosol uptake coefficient of $N_2O_5$ ($\gamma_{N2O5}$) and the ClNO2 yield ($\varphi_{ClNO2}$)."

5.  *Line 80. What is the "specific surface area"?*

It refers to the ratio of surface area concentrations to particle volume concentrations. To make it clear, we added a specific explanation in the corresponding text: "… and specific surface area (i.e. the ratio of surface area concentrations to particle volume concentrations)."

6. *Line 103. I found the phrase "and its chemical species" difficult to understand. Does this mean the chemical composition of the PM2.5?*

Yes, it means the chemical composition of the $PM_{2.5}$. To avoid confusion, we replaced "chemical species" with "chemical compositions" throughout the manuscript.

7. *Line 143. The ratio given in the equation is k2/k3 but its value is given as k3/k2. I think these should probably be given the same way up to avoid confusion.*

We have modified the equation by using the coefficient "$k_c$" instead of "$k_2/k_3$", and added the following description in line 170 – 171: "Where $k_c$ is the rate constant ratio representing the competition between aerosol-phase $H_2O$ and $Cl^-$ for the $H_2ONO_2^+(aq)$ intermediate and is fixed at 1/450 here, …".

8. *Line 152. The units of H2O, CL- and NO3 should be specified for completeness.*

Thanks for the comment. The units of $[H_2O]$, $[Cl^-]$ and $[NO_3^-]$ are mol $L^{-1}$. We have added the units of all variables throughout the text.

9. *Line 181. When the authors use the word estimate this is slightly confusing. If they have calculated the flux from these reactions its not an estimate it's a calculation. If they have found this from previous papers they should give the reference.*

Thanks for the comment. We have replaced "estimate" with "calculate" in the corresponding text as the results are calculated in the model.

10. *Line 197. The authors have defined acronyms for 4 areas in China. They don't use the acronyms that extensively. I would suggest they remove these acronyms*

*from the paper as it just confuses the reader who has to look back to the definition to find out what the areas are.*

We removed the acronyms and used full names for the 4 areas in China throughout the text.

11. *Line 310. The authors argue that more field measurements and model evaluations are needed for a more precise scaling factor. I would argue that they are needed not to come up with an improved scaling factor but to come up with an appropriate parameterisations which doesn't need a scaling factor at all.*

Thanks for the comment. We revised the corresponding text into "More field measurements and model evaluations are required to come up with a more precise parameterization better representing $\varphi_{ClNO2}$ in China."

12. *Line 318. The acronym CNEMC probably needs to be explained and more details provided of the data source.*

The full name and description of CNEMC are provided in line 302 – 305: "we also use observed hourly data of $O_3$ and $PM_{2.5}$ published by the China National Environmental Monitoring Center (CNEMC, http://www.cnemc.cn/sssj/, last access on June 20, 2021) to evaluate the model's overall performance in China. The network was launched in 2013 as part of the Clean Air Action Plan, and includes ∼1500 stations located in 370 cities by 2018 (Fig. S2)."
To make it clear, we also modified the text here into: "Figure 4 shows observed annual mean MDA8 $O_3$ and $PM_{2.5}$ in 2018 in China from CNEMC (China National Environmental Monitoring Center, introduced in Section 2.2) ..."

13. *Line 349. The authors argue that the increase in OH concentration is due to increased VOC oxidation. They have not provided any evidence for this. You might also expect the increased NOx concentration to lead to more OH through enhanced HO2+NO reactions, and the increased O3 concentration to lead to more primary OH production. Without a budget for the OH terms its no possible to attribute mechanism to the increased OH concentrations. Similarily its not*

*possible to attribute mechanism to the increased O3 concentration. Increases in NOx could lead to more O3 just as increased in VOC oxidation could.*

Thanks for the comment. We agreed with the reviewer that an increase in either $NO_x$ or VOCs could lead to more $O_3$. Therefore, as replied to the general comment #2, in addition to the map of changes in OH, we also added the map of changes in $HO_2$ between different simulation cases (Fig S7 (a)). We also revised the discussion here into: "The increased Cl atoms could react with VOCs (especially alkanes) producing more peroxy radicals, including organic peroxy radicals ($RO_2$) and hydroperoxyl radicals ($HO_2$). As shown in Figure S7 (a), the chlorine chemistry could increase annual mean $HO_2$ concentrations by $1.6 \times 10^6$ molec cm$^{-3}$ averaged in China (up to $8.6 \times 10^6$ molec cm$^{-3}$ in the coastal regions). In the presence of NO, the peroxy radicals recycle OH while oxidize NO to $NO_2$. The subsequent photolysis of $NO_2$ could further lead to more $O_3$ production and consequently also more OH (Osthoff et al., 2008; Riedel et al., 2014; Simpson et al., 2015). On the other hand, the recycling of $NO_x$ back into the atmosphere associated with the photolysis of $ClNO_2$ could also lead to more $O_3$ production. The results here show a significant increase in surface annual mean OH (Fig. S7 (b)) and MDA8 $O_3$ (Fig. 5c) by $3.8 \times 10^4$ molec cm$^{-3}$ and 1.1 ppbv respectively averaged in China (up to $1.2 \times 10^5$ molec cm$^{-3}$ and 4.5 ppbv respectively in the Sichuan Basin).".

14. *Line 353. The studies described here (Schmidt, Wang), suggest that Cl chemistry leads to reduced O3 concentration on global scale. The regional studies, notably in polluted regions highlight its importance in producing O3. This isn't clear in this paragraph and it is suggesting a disagreement in the literature that I don't think exists.*

Sorry for the confusion. We now revised the sentence into: "Both global and regional studies suggested that the heterogeneous $N_2O_5$ + Cl chemistry can enhance $O_3$ production through the production of Cl atoms and the recycling of $NO_x$ (Li et al., 2016; Sarwar et al., 2014; Wang et al., 2019).".

15. *Line 406. As mentioned earlier the attribution to the increased OH being due to increased VOC oxidation isn't supported by any evidence from the simulations. I would either perform a budget analysis on the ROx and OH production or soften*

*the language here to indicate that it may be due to these processes (Increased VOC oxidation, increased NOx leading to recuyling of HO2 into OH, or increased primary production from O3+hv).*

Thanks for the comment. As replied to the minor comment #13, we have added analysis on $HO_2$ in addition to OH. Accordingly, we also revised the text here into: "As discussed earlier in Section 3.2, increased Cl atoms could lead to more $HO_2$ and OH via VOCs oxidation. Combined with increased $NO_x$ associated with the release of $NO_2$ upon the photolysis of $ClNO_2$, further increases in both $O_3$ and OH could also be expected. The increase in OH is around $2 - 9 \times 10^4$ molec $cm^{-3}$ in central and eastern China on annual mean basis.".

16. *Line 460. I found the argument for the insensitivity of N2O5 to Chlorine emissions difficult to understand. Do the authors mean equation 1 or do they mean equation 2? Equation 1 doesn't really describe the gamma being used as presumabley the two terms have equations for describing the individual rates. I found the overall argument here difficult.*

Sorry for the confusion. We have added Eq. 2 and 3 for the calculation of $\gamma_{core}$ and $\gamma_{coat}$, respectively in the revised manuscript. More discussion is also provided to show the similarity and difference between different parameterizations. To make it clear, we also revised the text here into: "Unlike Yu parameterization, $N_2O_5$ concentrations have little dependence on chlorine emissions in McDuffie parameterization (Fig. 3a). This insensitivity to chlorine emissions could be expected from Eq. 2 where the dependence on aerosol chloride is not included so as to better reproduce wintertime reactive nitrogen observations in the eastern U.S.".

17. *Would it be useful to provide maps of the surface values for gamma N2O5 and the ratio of ClNO2 to HNO3 production for the different simulations run? How do these important values change spatially and between parameterizations?*

Thanks for the comment. We have added the maps of $\gamma_{N2O5}$, $\varphi_{ClNO2}$ and the ratios of $ClNO_2$ to $HNO_3$ for different simulation cases as Fig. S3 – S5 in the Supplementary Material. More discussion about the changes of $\gamma_{N2O5}$, $\varphi_{ClNO2}$, and $ClNO_2/HNO_3$ among different simulation cases are added in Section 3.1.

For example, in line 347 – 350 for the discussion of $\gamma_{N2O5}$ between the Base and McDuffie cases: "The overestimate of $N_2O_5$ in McDuffie parameterization suggests the potential underestimate in the corresponding $\gamma_{N2O5}$. As shown Figure S3, the value of $\gamma_{N2O5}$ from the McDuffie case is much smaller than that from the Base case (0.0071 vs. 0.016 averaged over China)."

For more detailed results, please see the discussion in Section 3.1 in the revised manuscript.